# Chemogenomics and orthology-based design of antibiotic combination therapies

Sriram Chandrasekaran[1,2,3,*], Melike Cokol-Cakmak[4], Nil Sahin[4], Kaan Yilancioglu[5], Hilal Kazan[6], James J Collins[2,3,7,8,9] & Murat Cokol[4,10,11,**]

## Abstract

Combination antibiotic therapies are being increasingly used in the clinic to enhance potency and counter drug resistance. However, the large search space of candidate drugs and dosage regimes makes the identification of effective combinations highly challenging. Here, we present a computational approach called INDIGO, which uses chemogenomics data to predict antibiotic combinations that interact synergistically or antagonistically in inhibiting bacterial growth. INDIGO quantifies the influence of individual chemical–genetic interactions on synergy and antagonism and significantly outperforms existing approaches based on experimental evaluation of novel predictions in *Escherichia coli*. Our analysis revealed a core set of genes and pathways (e.g. central metabolism) that are predictive of antibiotic interactions. By identifying the interactions that are associated with orthologous genes, we successfully estimated drug-interaction outcomes in the bacterial pathogens *Mycobacterium tuberculosis* and *Staphylococcus aureus*, using the *E. coli* INDIGO model. INDIGO thus enables the discovery of effective combination therapies in less-studied pathogens by leveraging chemogenomics data in model organisms.

**Keywords** chemogenomics; combination therapy; drug resistance; *Mycobacterium tuberculosis*; *Staphylococcus aureus*

**Subject Categories** Genome-Scale & Integrative Biology; Methods & Resources; Pharmacology & Drug Discovery

**Mol Syst Biol. (2016) 12: 872**

## Introduction

Antimicrobial combination therapy is utilized clinically to reduce treatment time, to prevent the emergence of resistance, and to expand the spectrum of pathogens targeted (Dancey & Chen, 2006; Hopkins, 2008; Fischbach & Walsh, 2009; Bush *et al*, 2011; Worthington & Melander, 2013). However, individual drugs in combination can enhance or interfere with other drugs' actions leading to unexpected effects. Therefore, mapping antibiotic interactions is essential for designing effective combination therapies (Dancey & Chen, 2006; Nichols *et al*, 2011; Roemer & Boone, 2013; Worthington & Melander, 2013). Systematic screens for drug interactions require testing various ratios of the component drugs (Hopkins, 2008), and the experimental identification of combinations with robust effects across a broad dose range is resource- and time-intensive, even for a small number of compounds (Zimmermann *et al*, 2007). The large space of candidate drugs, their dosage and chemical properties, and the nature of the target pathogen all affect the choice of the drug combination, making it a challenging and complex problem (Dancey & Chen, 2006; Hopkins, 2008). Because the number of possible drug combinations is astronomically high, new *in silico* methods are crucial for determining the most promising combinations (Hopkins, 2008; Lehar *et al*, 2008). This mandates the integration of genomic and chemical data using a computational framework. An integrated computational platform for predicting antibiotic interactions could pave the way for the rapid assessment of novel antimicrobial combinations before entry into clinical usage.

Here, we present an approach entitled *INferring Drug Interactions using chemo-Genomics and Orthology* (INDIGO), which predicts antibiotic combinations that interact synergistically or antagonistically in inhibiting bacterial growth based on the chemogenomic profiles of the individual antibiotics. Chemogenomic profiling measures fitness of gene-knockout strains treated with bioactive compounds

1 Harvard Society of Fellows, Faculty of Arts and Sciences, Harvard University, Cambridge, MA, USA
2 Broad Institute of MIT and Harvard, Cambridge, MA, USA
3 Wyss Institute for Biologically Inspired Engineering, Harvard University, Cambridge, MA, USA
4 Faculty of Engineering and Natural Sciences, Sabanci University, Istanbul, Turkey
5 Department of Molecular Biology and Genetics, Uskudar University, Istanbul, Turkey
6 Department of Computer Engineering, Antalya International University, Antalya, Turkey
7 Department of Biological Engineering, Institute for Medical Engineering & Science, Massachusetts Institute of Technology, Cambridge, MA, USA
8 Synthetic Biology Center, Massachusetts Institute of Technology, Cambridge, MA, USA
9 Harvard-MIT Program in Health Sciences and Technology, Cambridge, MA, USA
10 Department of Molecular Biology and Microbiology, Tufts University School of Medicine, Boston, MA, USA
11 Laboratory of Systems Pharmacology, Harvard Medical School, Boston, MA, USA
 *Corresponding author. Tel: +1 617 496 0048; E-mail: chandrasekaran@fas.harvard.edu
 **Corresponding author. Tel: +1 617 432 6164; E-mail: murat_cokol@hms.harvard.edu

and provides unbiased insights into the mechanism of action of drugs (Bredel & Jacoby, 2004; Lehar *et al*, 2008; Ho *et al*, 2009; Nichols *et al*, 2011; Lee *et al*, 2014). We developed and utilized INDIGO to analyze a large compendium of publicly available chemogenomic data in *Escherichia coli* (Nichols *et al*, 2011) to identify predictive genetic features of antibiotic synergy and antagonism, and subsequently infer novel drug interactions. By finding orthologs of genes identified by INDIGO to be predictive of drug interactions in *E. coli*, we successfully predicted drug synergy and antagonism in the bacterial pathogens *Mycobacterium tuberculosis* and *Staphylococcus aureus*. These two pathogens cause a significant mortality worldwide and are frequently treated clinically using combinations of antibiotics (Liu *et al*, 2011; Ramon-Garcia *et al*, 2011). INDIGO greatly expands the capability of current drug-interaction prediction approaches by estimating the interaction outcomes in pathogens using chemogenomic data from model organisms.

# Results

## Experimental measurement of 105 interactions among 15 drugs as training data

To train INDIGO, 15 compounds with available chemogenomic profiles were selected (Nichols *et al*, 2011) (Table 1). These compounds, consisting of 14 antibiotics and a stress agent (henceforth referred to as drugs), covered a variety of classes including cell wall, transcription, translation, and DNA metabolism inhibitors. The interaction outcome of all pairwise combinations of these 15 drugs (105 pairs) was measured in duplicate. For each drug pair, individual drugs were combined in a 4 × 4 2-dimensional checkerboard assay, with each drug's dose linearly increasing from 0 to near minimum inhibitory concentration (MIC) in each axis (Fig 1; Table 1; Appendix Fig S1; Materials and Methods).

The Loewe additivity model was used to quantify the drug interactions (Loewe, 1953). The Loewe additivity model defines a drug as additive or noninteracting with itself. Antagonistic and synergistic interactions are inferred based on the deviations from additivity. Synergy implies that the same amount of growth inhibition is achieved with a lower dose when both drugs are combined. For the Loewe model, we used as input the growth of the bacteria at different dose combinations of the individual drugs to generate a quantitative interaction score ($\alpha$ score) between the drug pair, with low scores ($\alpha < -0.5$) corresponding to synergy and high scores ($\alpha > 1$) corresponding to antagonism (Fig 1; Dataset EV1; Materials and Methods). This sensitive framework approximately corresponds to a bliss interaction score of $-0.3$ for $\alpha < -0.5$ and $+0.3$ for $\alpha > 1$ and provides a robust measure of drug interaction across doses (Appendix Fig S2). The measured interaction scores had strong correlation between the two biological replicates (rank correlation ($R$) = 0.81, *P*-value = $10^{-26}$) and were consistent with existing interaction data (Appendix Fig S3; Appendix Supplementary Methods). Overall, among the 105 drug combinations, antagonistic drug interactions were more common than synergistic interactions. Further,

**Table 1. List of drugs used in this study and their targets.**

| Compounds | Abbreviation | Target process | Symbol | Drug class | MIC (μg/ml) |
|---|---|---|---|---|---|
| Amikacin | AMK | Protein synthesis, 30S | | Aminoglycoside | 4 |
| Gentamicin | GEN | Protein synthesis, 30S | | Aminoglycoside | 3 |
| Tobramycin | TOB | Protein synthesis, 30S | | Aminoglycoside | 2.5 |
| Tetracycline | TET | Protein synthesis, 30S | | Tetracycline | 18 |
| Chloramphenicol | CHL | Protein synthesis, 50S | | Phenylpropanoid | 5 |
| Clarithromycin | CLA | Protein synthesis, 50S | | Macrolide | 22 |
| Erythromycin | ERY | Protein synthesis, 50S | | Macrolide | 13 |
| Ciprofloxacin | CIP | DNA gyrase | | Quinolone | 0.01 |
| Levofloxacin | LEV | DNA gyrase | | Quinolone | 0.013 |
| Nalidixic acid | NAL | DNA gyrase | | Quinolone | 5 |
| Trimethoprim | TRI | Folic acid biosynthesis | | Pyrimidine | 0.35 |
| Oxacillin | OXA | Cell wall | | Beta-lactam | 190 |
| Cefoxitin | CEF | Cell wall | | Beta-lactam | 4.5 |
| H₂O₂ | H22 | Oxidative stress | **ROS** | Stress | 250 |
| Nitrofurantoin | NIT | Multiple mechanisms | | Furan | 12 |
| *Fusidic acid* | FUS | Elongation factor—protein synthesis | | Fusidane | 800 |
| *Rifampicin* | RIF | RNA synthesis | | Rifampin | 2.5 |
| *Vancomycin* | VAN | Cell wall | | Glycopeptide | 180 |
| *Spectinomycin* | SPE | 30S protein synthesis | | Aminocylitol | 5.6 |

The antibiotics used in the test set for validation are highlighted in italics.

    

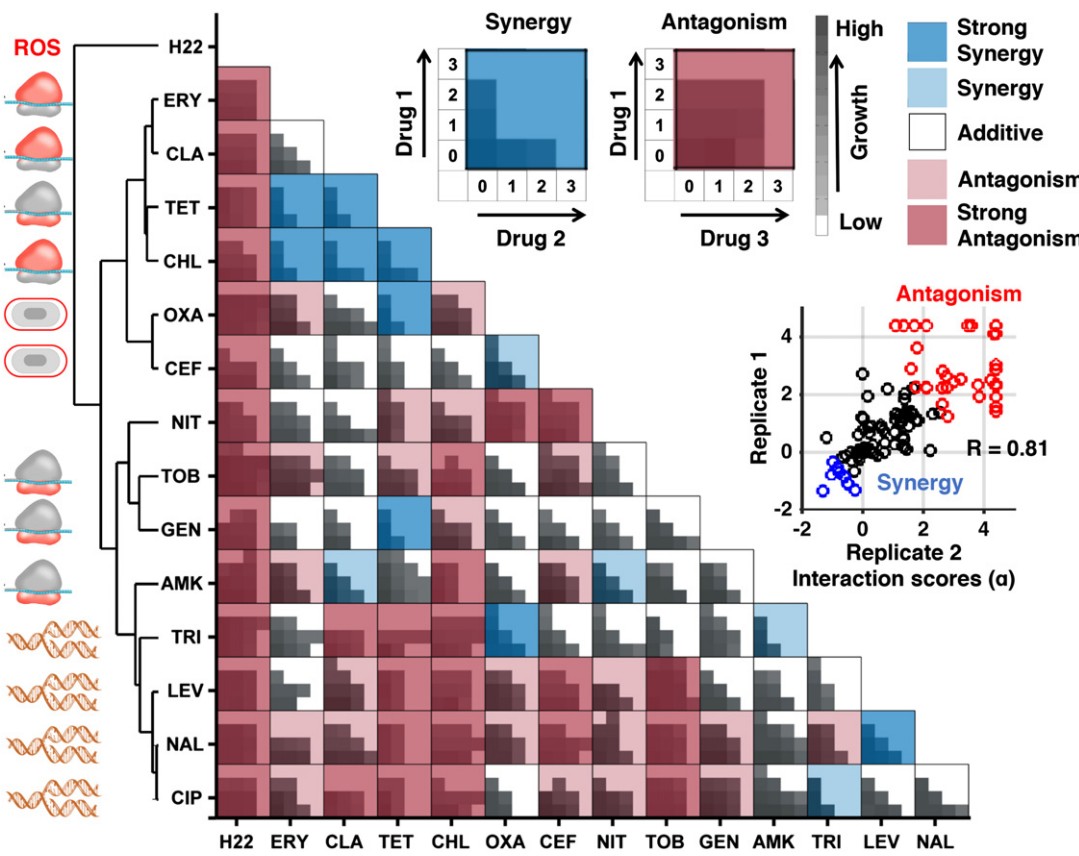

**Figure 1.  Experimental drug-interaction data for training INDIGO.**
Shown are growth data of *Escherichia coli* strains grown in the presence of different antibiotic combinations. Each drug-interaction experiment consisted of 16 growth measurements in various dose combinations of each antibiotic, as described in Materials and Methods. For each drug pair, the growth rates were measured for all pairwise combinations of four drug concentrations, linearly increasing from 0 to the minimum inhibitory concentration (MIC). The area under the growth curve is plotted for each drug pair at different doses. The average of two replicates is shown. These data were used as input to the Loewe's model to calculate drug-interaction scores, which are overlaid on the growth data. The Loewe's model outputs a quantitative score ($\alpha$-score) for each drug-interaction measurement. The $\alpha$-scores were calculated for each replicate and the average score was overlaid on the growth data. The inset shows strong rank correlation between the alpha scores for the two biological replicates. The interaction scores were discretized for visual clarity. The following classes were used: strong synergy ($\alpha$ score $< -0.5$), mild synergy ($\alpha$ score $< -0.25$), strong antagonism ($\alpha$ score $> 2$), and antagonism ($\alpha$ score $> 1$). The antibiotics were clustered based on interaction score profile similarity. The target process of each antibiotic is depicted next to its abbreviation and is described in Table 1.

unsupervised clustering of interactions grouped the drugs based on their mechanism of action, as well as their bacteriostatic and bactericidal properties, consistent with previous studies (Yeh *et al*, 2006; Ocampo *et al*, 2014). In sum, these observations support the accuracy and robustness of the drug-interaction data used for training INDIGO.

**Framework for predicting drug–drug interactions using chemogenomic profiles**

The training data (input) for INDIGO consists of (i) chemogenomic profiles of individual drugs of interest and (ii) experimental interaction scores for the combinations of drugs. A chemogenomic profile of a drug is an array of fitness scores for gene-deletion strains treated with the drug of interest compared to the wild-type strain. We transformed the chemogenomic profile of a drug into a binary sensitivity profile by identifying the deletion strains that are significantly sensitive to a drug (Materials and Methods). The sensitivity

profiles of individual drugs in a drug combination are combined by INDIGO using Boolean operations to create a joint profile. In this framework, the union (sigma score) and intersection (delta score) operations capture similarity and uniqueness in the mechanism of action of the individual drugs. A machine learning algorithm called random forests is then used to build a predictive model that links the interaction outcome of drug combinations to the joint chemogenomic profile of the drug pair (Materials and Methods; Fig 2). The random forest algorithm builds an ensemble of decision trees using the training dataset and outputs the mean prediction of the individual trees; it also identifies genes in the chemogenomics data that are most predictive of drug interactions. INDIGO learns the mechanism of drug interactions from the chemogenomics data in an unbiased fashion by using the random forest algorithm.

INDIGO thus takes a systems approach to predict drug interactions; the approach quantifies the contribution of individual genes with a chemical–genetic interaction on the overall drug-interaction

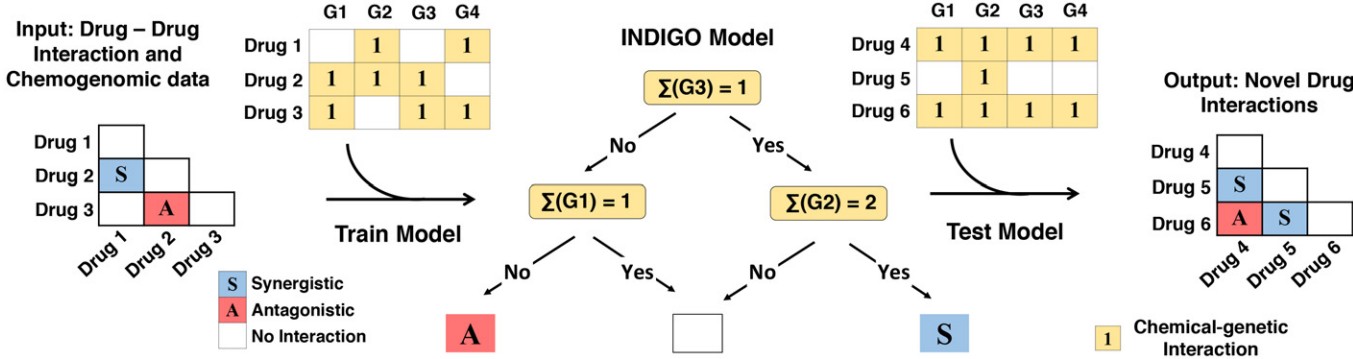

**Figure 2. Overview of the INDIGO approach to study drug interactions.**
INDIGO takes in as input drug-interaction data measuring synergistic and antagonistic interaction outcomes, and a compendium of chemogenomic profiles that identify gene deletions that enhance sensitivity to a compound of interest. INDIGO subsequently integrates the individual drug chemogenomic profiles using Boolean operations (sigma and delta scores) that capture similarity and dissimilarity (only sigma scores are shown for simplicity). INDIGO identifies top genomic predictors of drug synergy and antagonism using a random forest algorithm. The inferred model is then used to predict interaction outcomes for combinations of drugs with known chemogenomic profiles.

outcome (synergy or antagonism). Importantly, the contribution of each gene in INDIGO is contextual; that is, it depends on the state of other genes. We trained INDIGO using our experimental pairwise drug–drug interaction data along with chemogenomic profiles for these 15 drugs from Nichols *et al* (Nichols *et al*, 2011), which screened 73 drugs and 53 stress agents against 3,979 nonessential gene-deletion strains in *E. coli*. This trained INDIGO model can predict the pairwise interactions of any two drugs with known chemogenomics profiles.

## Experimental validation of novel predictions for 66 drug–drug interactions

In order to assess the predictive power of INDIGO, a test set of four additional antibiotics whose chemogenomic profiles were reported in Nichols *et al* were chosen. These antibiotics had distinct chemical properties and mechanisms of action compared to the antibiotics used in the training set. Specifically, we tested rifampicin (which targets RNA polymerase), vancomycin (a glycopeptide that targets the cell wall), fusidic acid (which targets the elongation factor in protein synthesis), and spectinomycin (an aminocylitol drug that targets the 30S ribosome subunit similar to aminoglycoside antibiotics). Using INDIGO, interaction scores were predicted for 66 drug pairs involving 15 drugs in the training set and these four new antibiotics. We then experimentally measured interaction scores for these pairs using the same experimental setup as the training set.

Among the top ten synergistic and antagonistic predictions by INDIGO, six synergistic and seven antagonistic predictions were experimentally validated, respectively (Fig 3B; Appendix Table S1). INDIGO correctly predicted two synergistic interactions comprising antibiotics that were both not in the training set (fusidic acid–rifampicin, fusidic acid–vancomycin). Figure 3C demonstrates the significant correlation between the experimental and predicted drug-interaction scores for all 66 drug pairs in the test set ($R = 0.52$, $P$-value = $10^{-6}$). These results show that INDIGO can successfully predict interactions among drugs with known chemogenomic profiles.

To benchmark INDIGO, the predictions were compared to a chemogenomics-based approach used to infer antifungal interactions, which predicted synergy based on chemogenomic profile similarity of the individual drugs (Jansen *et al*, 2009; Materials and Methods). In this approach, the similarity of two drugs was evaluated by measuring the Pearson's correlation between the individual drugs' chemogenomic profile, or through the total overlap between the genes present in the chemogenomic profile of individual drugs. These approaches performed poorly in predicting experimental interaction scores compared to INDIGO ($R = 0.14$, $P$-value = 0.25, Fig 3D–F; Appendix Fig S4). INDIGO also significantly outperformed the O2M algorithm (Brown *et al*, 2014), which is an extension of the Jansen *et al*'s overlap-based approach (Appendix Fig S5).

Similarity and overlap-based approaches are less effective in predicting interaction outcomes for new classes of drugs and lack a model for antagonism. Thus, taking into account the identity of the individual genes in the chemogenomic profile and accounting for both similarity and dissimilarity increases the ability to predict drug interactions by INDIGO. Our data show that drugs with similar targets and chemogenomic profiles can have both synergistic and antagonistic outcomes. For example, combinations of tobramycin–spectinomycin, tobramycin–gentamicin, and fusidic acid–clarithromycin share similar chemogenomic profiles and target processes, yet have antagonistic, neutral, and synergistic outcomes, respectively (Fig EV1).

Importantly, INDIGO quantitatively predicts drug interactions and its predictive ability was not influenced by any specific threshold for inferring synergy or antagonism (Appendix Fig S6), it was robust to the choice of methods used to evaluate predictions (Materials and Methods; Appendix Fig S4), and its predictions were more likely to be synergistic than a random screen ($P < 10^{-19}$; *t*-test; Appendix Fig S7). Further, the results were robust to the removal of the outlier compound hydrogen peroxide that had promiscuous antagonistic interactions (Appendix Fig S8), and led to similar accuracy based on cross-validation analysis (Appendix Figs S9, S10, S11 and S12). Cross-validation analysis also revealed that predictions were consistently accurate across most drugs and identified

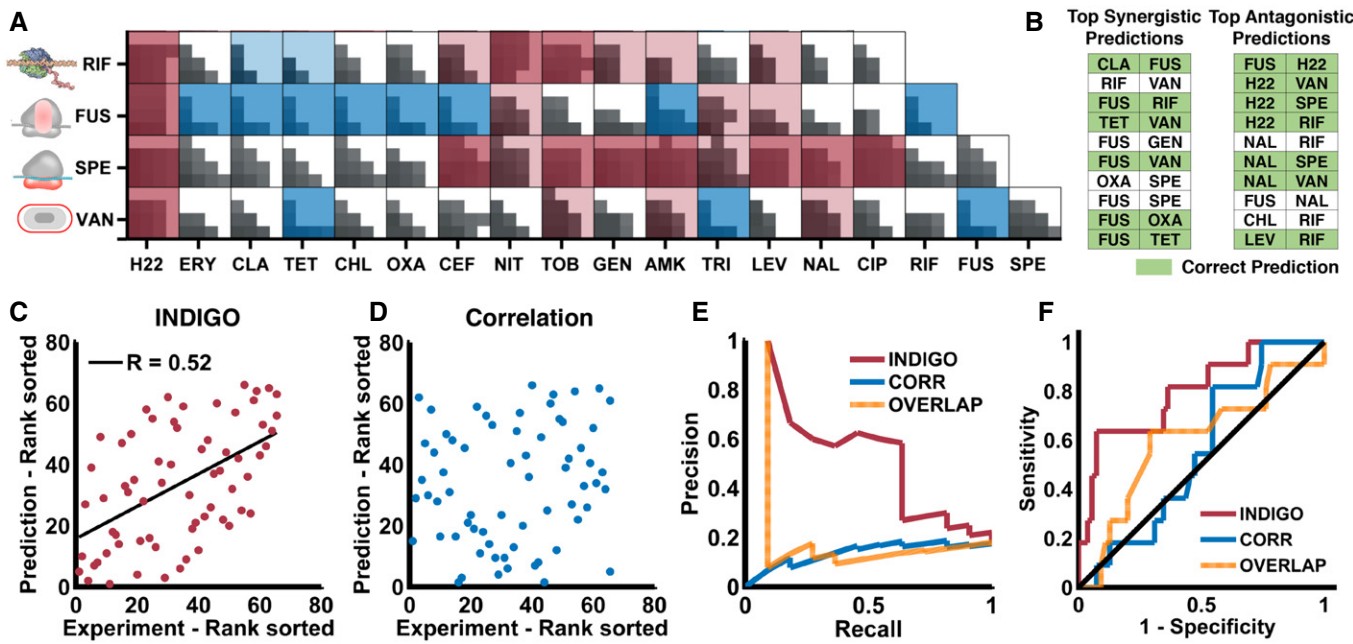

**Figure 3.  INDIGO accurately predicts drug interactions.**

A   INDIGO was evaluated by its ability to predict drug-interaction outcomes in new test data. Experimental interaction data for 66 pairwise interactions between four new drugs that were chosen for testing are shown.

B   Table shows the top ten most synergistic and antagonistic predictions by INDIGO. See Appendix Table S1 for the entire list of predictions.

C   Scatter plot of measured interaction scores and predictions by INDIGO ($R = 0.52$, $P$-value $= 10^{-6}$).

D   Scatter plot showing predicted interaction scores by chemogenomic profile similarity-based approach ($R = 0.14$, $P$-value $= 0.25$; Materials and Methods).

E   Precision (fraction of identified interactions that are true positives) and recall (fraction of true positives interactions correctly identified) for INDIGO (shown in red) in predicting synergistic drug interactions. Analogous curves for correlation (blue)- and profile overlap (orange)-based approaches are shown for comparison. INDIGO achieves at least three times higher precision than other approaches over a range of recall levels. See Appendix Fig S4 for analogous curves for antagonism.

F   Sensitivity (true-positive rate) and specificity (true-negative rate), measured over a range of thresholds for predicting synergistic interactions in the validation data. See Appendix Fig S4 for analogous curves for antagonism.

Source data are available online for this figure.

hydrogen peroxide as leading to the largest errors in predictions (Appendix Figs S8 and S9). This suggests that the chemogenomic profile of hydrogen peroxide may not accurately reflect its mechanism of interaction. In sum, INDIGO successfully predicts novel drug interactions as assessed by several metrics.

**Global interaction landscape of compound combinations**

To further strengthen our model, INDIGO was re-trained on the entire experimental drug–drug interaction dataset, including the test-set interactions (171 drug pairs). This model was then applied to the entire compendium of 73 drugs and 53 stress-agent chemogenomic profiles from Nichols *et al.* By mining this dataset (2,628 drug–drug interactions and 3,869 drug–stress agent interactions), we observed the well-known antagonistic drug interactions between bactericidal antibiotics targeting DNA synthesis (such as ciprofloxacin and other quinolones) and bacteriostatic antibiotics targeting translation (such as tetracycline) (Ocampo *et al*, 2014; Lobritz *et al*, 2015). Our interaction dataset also included nonantibiotic drugs that could be repurposed as antibiotic adjuvants (Gill *et al*, 2015). Interestingly, membrane stress agents, such as the detergents sodium dodecyl sulfate (SDS) and triclosan, were found to be synergistic with a large number of antibiotics (Fig 4). These

predictions are consistent with a previous high-throughput drug combination screen in *E. coli* (Farha & Brown, 2010).

INDIGO correctly predicted synergy between commonly used, clinically validated antibiotic combinations. Vancomycin was synergistic with beta-lactam antibiotics (ampicillin and aztreonam), as was the combination of ampicillin and gentamicin, which has been a mainstay combination treatment for intra-abdominal infections (Solomkin *et al*, 2010; Liu *et al*, 2011). Rifampicin, which is frequently used in combination with other antibiotics, particularly in the case of biofilm-associated infections, is predicted to be synergistic with oxacillin, macrolide, and tetracycline antibiotics, affirming the clinical efficacy of this drug with other companion antibiotics (Forrest & Tamura, 2010). Our approach thus consistently and rapidly identifies synergy among antibiotic combinations that required clinical validation over decades of study.

**Genetic predictors of synergy and antagonism**

Surprisingly, a small set of genes account for most of the predictive ability of our model. Genes were assigned an importance score by INDIGO proportional to their relative contribution in predicting drug interactions. The top 81 genes accounted for 50%, the top 222 accounted for 75%, and the top 581 accounted for 95% of variance

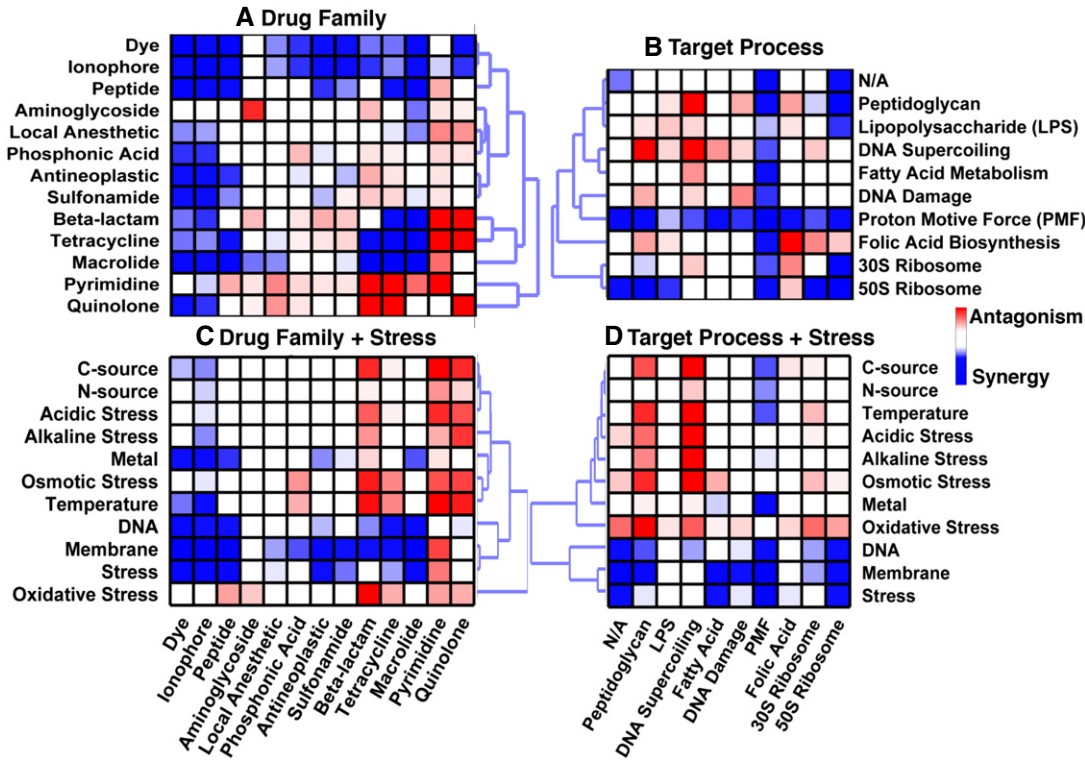

**Figure 4.  Global analysis of drug–drug and drug–stress interactions.**
Panels A and B summarize the interaction outcome between drugs from similar or different families and target processes. The heat maps represent the interactions inferred from the entire compendium of 73 drugs and 53 stress-agent chemogenomic profiles from Nichols *et al*, collapsed into 13 major drug families (panel A) or 10 major target processes (panel B). To determine average synergistic or antagonistic interactions between different groups, we compared the distribution of interaction scores for all drug combinations between two groups with the background interaction score for all drug pairs (Appendix Supplementary Methods). Panels C and D summarize the interaction outcomes between drugs and stress agents grouped based on drug family (panel C) or target process (panel D). N/A—target process not available, based on annotation from Nichols *et al*.

in the predicted data (Fig EV2; Dataset EV3). We defined the set of top 250 genes that contributed to over 75% of the variance predicted by the model as drug interaction-related (DIR) genes.

The presence of these DIR genes in the sensitivity profile of the drugs also correlated with either synergy or antagonism (Table 2A). Overall, the interaction outcomes for each drug combination depend on complex interactions between many genes. Nevertheless, the presence of these top genes can be a strong predictor of synergy or antagonism. For example, we find genes associated with synergy to be enriched in the chemogenomic profile of triclosan ($P$-value = $10^{-10}$, hypergeometric test), explaining its promiscuous synergy. By ranking genes used by INDIGO, we can determine the dominant pathways that are commonly used across drug classes. KEGG pathway enrichment analysis among the DIR genes identified processes that are targeted by the drugs, and pathways known to be involved in mediating drug interactions and resistance such as efflux pumps (Webber & Piddock, 2003) (Table 2B). Interestingly, in addition to known targets of antibiotic action—translation, cell wall synthesis, DNA replication, folic acid biosynthesis, and RNA metabolism—genes related to central energy metabolism and drug transport were also enriched in the DIR genes. Notably, lipopolysaccharide biosynthesis and oxidative phosphorylation were found as two of the top ten pathways associated with drug interactions. This

is in agreement with a recent study that measured antibiotic interactions in all nonessential gene-deletion strains of *E. coli* and found that strains harboring deletions in these pathways resulted in altered interaction outcomes (Chevereau & Bollenbach, 2015). This further affirms the validity and significance of the DIR genes identified by INDIGO.

**Predicting drug interactions across species using orthology mapping**

While chemogenomic profiling data are widely available for *E. coli*, such datasets are often not available for other microbes, including many pathogens. We hypothesized that INDIGO could be applied to clinically relevant pathogens by identifying orthologs of *E. coli* DIR genes. We tested this hypothesis using the Gram-positive pathogen *S. aureus*. Genes that were orthologous between *E. coli* and *S. aureus* were mapped onto the INDIGO model (Materials and Methods). Interestingly, the top 250 DIR genes in the model were significantly enriched for genes orthologous between *E. coli* and *S. aureus* ($P$-value = 0.001, hypergeometric test; total orthologs = 844 genes). This observation suggests that interaction outcomes in *S. aureus* can be predicted using the *E. coli* INDIGO model. We hypothesized that interactions that depend on orthologous genes

**Table 2. A core set of genes and pathways are predictive of drug interactions.**

| Gene | Gene name | Rank | Associated with | P-value (t-test) Sensitive in One drug | Both drugs |
|------|-----------|------|------------------|------------|------------|
| (A) | | | | | |
| glmS | L-Glutamine: D-fructose-6-phosphate aminotransferase | 1 | Synergy | 0.13 | $10^{-6}$ |
| mdtK | Multidrug efflux protein | 2 | Synergy | $10^{-3}$ | 0.17 |
| greA | Transcription elongation factor | 3 | Antagonism | $10^{-7}$ | $10^{-5}$ |
| asmA | Putative assembly protein | 4 | Synergy | $10^{-5}$ | $10^{-5}$ |
| prpC | Methylcitrate synthase | 5 | Antagonism | $10^{-11}$ | 0.31 |
| yfiN | Predicted diguanylate cyclase | 6 | Antagonism | $10^{-11}$ | 0.31 |
| cyaY | Frataxin, iron-binding and oxidizing protein | 7 | Synergy | 0.05 | 0.01 |
| cspE | Cold shock protein | 8 | Antagonism | $10^{-10}$ | 0.09 |
| rfbX | Predicted polisoprenol-linked O-antigen transporter | 9 | Antagonism | $10^{-5}$ | $10^{-3}$ |
| yphF | Putative LACI-type transcriptional regulator | 10 | Synergy | 0.12 | 0.05 |

| Pathway | P-value | FDR | Associated with | Similar or dissimilar between drugs |
|---------|---------|-----|------------------|--------------------------------------|
| (B) | | | | |
| Homologous recombination | $10^{-5}$ | 0.02 | Antagonism | Dissimilar |
| Ribosome | $10^{-4}$ | 0.02 | | |
| Purine metabolism | $10^{-4}$ | 0.02 | Synergy | |
| Mismatch repair | $10^{-4}$ | 0.04 | Antagonism | Dissimilar |
| Lipopolysaccharide biosynthesis | $10^{-4}$ | 0.04 | Synergy | Similar |
| Drug resistance | $10^{-4}$ | 0.04 | Synergy | Similar |
| Pentose phosphate pathway | $10^{-3}$ | 0.04 | | |
| Alanine, aspartate, and glutamate metabolism | $10^{-3}$ | 0.04 | Synergy and Antagonism | |
| Pyrimidine metabolism | $10^{-3}$ | 0.09 | Synergy | |
| Bacterial chemotaxis | $10^{-3}$ | 0.09 | | |
| Bacterial secretion system | $10^{-3}$ | 0.09 | | |
| Peptidoglycan biosynthesis | 0.02 | 0.19 | | |
| Oxidative phosphorylation | 0.03 | 0.22 | Synergy and Antagonism | Dissimilar |

(A) Top 10 genes most predictive of antibiotic interactions inferred by INDIGO and their correlation with synergy or antagonism across all drug interactions are shown. Further, the presence of these genes in either one (dissimilar) or both (similar) the antibiotics' chemogenomic profile led to different effects on the interaction outcome. For example, the top predictor gene, glmS, is significantly associated with synergy when present in the profile of both the drugs (P-value = $10^{-6}$), but not when present in only one of the drugs in a drug pair (P-value = 0.13). (B) Top 10 KEGG pathways enriched among the top 250 DIR genes. The enrichment of these pathways among genes that were either significantly (P-value < 0.05; FDR < 0.25) associated with synergy or antagonism is displayed in columns 2 and 3. For example, the homologous recombination pathway was significantly enriched among the top 250 genes (P-value = $10^{-5}$). In addition, this pathway was significantly enriched among genes that were strongly associated with antagonism (P-value < 0.05) when present in one of the drugs in a drug pair (dissimilar). Empty boxes indicate no significant enrichment toward synergy or antagonism.

would be conserved between the two species, while those that are predicted by INDIGO to depend on non-orthologous genes would not be conserved. Therefore, we used the *E. coli* chemogenomic profiles of the orthologous genes to predict drug-interaction scores for *S. aureus* (Materials and Methods). To quantify the relative contribution of non-orthologous genes on each drug-interaction outcome, the chemogenomic scores of non-orthologous genes in the *E. coli* INDIGO model were set to take minimal value (Materials and Methods). This process corresponds to deleting these genes from the *E. coli* INDIGO model. The predictions from this modified model were then compared with the original *E. coli* model predictions. The

difference in scores for each interaction, called the deviation score, allowed us to identify drug interactions most sensitive to the state of the non-orthologous genes and therefore less likely to be conserved between *E. coli* and *S. aureus*.

We predicted interaction and deviation scores for all 171 drug combinations that were experimentally tested in *E. coli* using this orthology framework. To experimentally validate the predictions in *S. aureus*, 10 drugs from the *E. coli* dataset were chosen; these drugs belong to different classes that were predicted by INDIGO to have the greatest range of variation in their interactions between *E. coli* and *S. aureus* (Materials and Methods; Fig 5, Appendix Fig

S13, and Appendix Tables S2 and S3). The interaction score for all 45 pairs of these antibiotics was then experimentally measured in *S. aureus* using the same setup used for *E. coli* (Fig 5B). Overall, we found that there is a small yet significant correlation between the experimentally measured drug-interaction scores of the two species ($R = 0.39$; *P*-value = 0.008, Appendix Fig S14). This is in agreement with the observation that the top predictive genes are conserved between the two species. Consistent with our hypothesis, interactions that are associated with non-orthologous genes as quantified by the deviation score have stronger differences in their interaction outcome between the two species ($R = 0.52$; *P*-value = $10^{-4}$; Appendix Table S4; Fig 5C). Our analysis indicates that although some drug interactions are conserved between the two species, there are many exceptions. These exceptions can be identified by INDIGO based on the deviation score. By taking into account these deviations, interaction outcomes in *S. aureus* can be directly estimated using INDIGO ($R = 0.47$ between experimental and predicted *S. aureus* interactions; *P*-value = $10^{-3}$; Figs 5D and EV3). For example, INDIGO correctly discovered that the oxacillin–tetracycline combination is synergistic in *E. coli,* but antagonistic in *S. aureus*. These results are noteworthy given that *S. aureus* interaction data

were not used for training the model, and chemogenomics data from a different species were used to predict interactions.

To further validate the applicability of our approach to other organisms, INDIGO was applied to predict antibiotic interaction outcomes in the pathogen *Mycobacterium tuberculosis* using *E. coli* chemogenomics data; 909 genes in *M. tuberculosis* were orthologous with genes in our *E. coli* dataset. Again, these conserved genes are enriched among the top 250 DIR genes (*P*-value = $10^{-4}$, hypergeometric test). Using the INDIGO *M. tuberculosis* model, interaction outcomes were predicted for 24 antibiotic combinations available in the literature with chemogenomic data in our compendium (Bhusal *et al*, 2005; Ramon-Garcia *et al*, 2011; Singh *et al*, 2015). Similar to *S. aureus* predictions, we found strong correlation between the predicted and experimentally observed interaction outcomes in *M. tuberculosis* ($R = 0.54$; *P*-value = 0.006; Figs 5E and EV3; Appendix Table S5). Many combinations that are synergistic in *E. coli* are also predicted to be synergistic in *M. tuberculosis*. For example, combinations of spectinomycin with clarithromycin or azithromycin are synergistic in both species. Remarkably, 20 out of the 24 antibiotic combinations for which predictions were made for *M. tuberculosis* were not in our experimental dataset for *E. coli* and

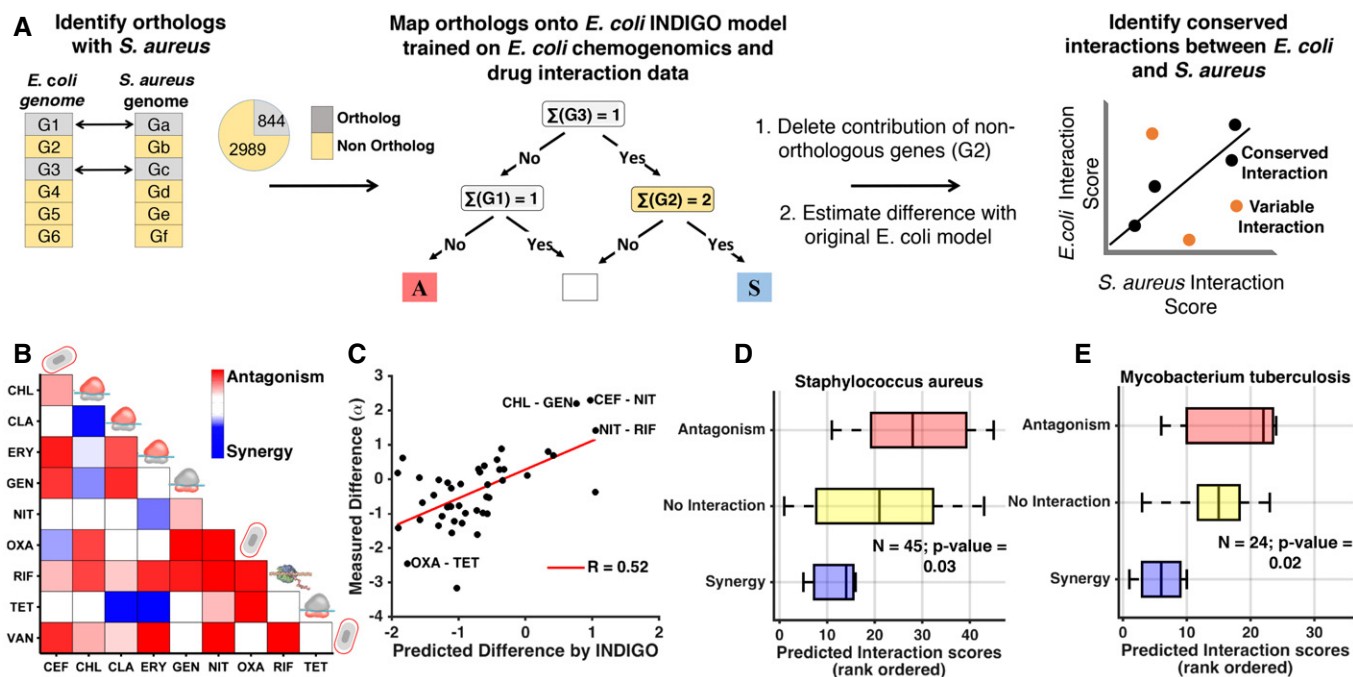

**Figure 5.    Predicting drug-interaction outcomes in *Staphylococcus aureus* and *Mycobacterium tuberculosis* using INDIGO.**

A    Overview of our framework for predicting drug-interaction data in *S. aureus* using *Escherichia coli* drug-interaction and chemogenomics data. The model predicts interactions that would differ between the two species.

B    Experimental *S. aureus* drug-interaction data. Average interaction scores of two replicates are shown. The target process of the antibiotics is depicted as cartoons and is described in Table 1.

C    Scatter plot of predicted difference in interaction score between the two species and experimentally measured difference in the interaction score. Interactions that have the largest differences in interaction scores between the two species are highlighted.

D    Box plots show predicted interaction score by INDIGO for experimentally measured synergistic and antagonistic interactions in *Staphylococcus aureus* ($R = 0.47$; *P*-value = 0.001; one-way ANOVA *P*-value = 0.03).

E    Box plots show predicted interaction score by INDIGO for synergistic and antagonistic interactions in *Mycobacterium tuberculosis* ($R = 0.54$; *P*-value = 0.006; one-way ANOVA *P*-value = 0.02). *Mycobacterium tuberculosis* interaction data were curated from literature and are qualitative in nature. See Fig EV3 for a more detailed plot with drug labels.

Source data are available online for this figure.

    

hence could not have been predicted from experimental data directly.

We have thus established that we can reliably estimate drug-interaction outcomes in *E. coli*, *S. aureus,* and *M. tuberculosis*. To assess the organism-specific variability of interactions, the entire compendium of interactions between 73 drugs and 53 stress agent from Nichols *et al* was compared across the three species (Dataset EV2). Combinations involving tetracyclines and quinolones were found to be more likely to vary across species (Fig EV4). Consistent with this observation, no significant enrichment was found for orthologous genes among the DIR genes in the chemogenomic profile of these antibiotics (*P*-value > 0.05, hypergeometric test). Among all predictions, 3 and 5% were predicted to exhibit strong broad-spectrum synergy and antagonism in all three species, respectively. In contrast, only 1% of the drug pairs showed a mix of synergy and antagonism in the three species.

## Discussion

Despite an alarming rise in resistance, few new antibiotics have been developed in the past decade, necessitating the increased use of antibiotic combination therapies. In this study, we developed and validated a computational framework that can successfully predict synergistic and antagonistic drug interactions in three bacterial species. In addition to the 171 drug pairs experimentally assessed in *E. coli*, drug-interaction predictions were validated for 45 drug pairs in *S. aureus*. In agreement with previous studies in bacteria and yeast, antagonistic interactions were more prevalent than synergistic interactions. This underscores the difficulty of selecting drug combinations that do not adversely affect clinical outcomes.

An important finding from this study is that drug interactions can be predicted based on chemogenomic profiles of individual drugs. This suggests that cellular sensitivity to drug combinations can be abstracted by a linear model of individual drug-sensitivity profiles (Pritchard *et al*, 2013). This is consistent with the observation that protein and transcriptional dynamics of cells treated with drug combinations are a linear combination of the individual drug effects (Geva-Zatorsky *et al*, 2010; Chevereau & Bollenbach, 2015). A large resource of chemogenomic profiling is now available for *E. coli*, *S. cerevisiae*, and human cancer cell lines; INDIGO can thus be applied to a wide range of drugs across different systems (Flores *et al*, 2005; Tamae *et al*, 2008; Girgis *et al*, 2009; Ho *et al*, 2009; Muellner *et al*, 2011; Nichols *et al*, 2011; Lee *et al*, 2014).

INDIGO assesses the influence of hundreds of individual chemical–genetic interactions on drug–drug interaction outcomes. In contrast, existing chemogenomics-based approaches determine synergy based on a single aggregate metric of drug similarity. Our gene-centric model has also enabled us to apply our *E. coli* drug-interaction model to predict outcomes in other bacterial species. Genes identified to be predictive of drug interactions by INDIGO (DIR genes) were involved in the processes targeted by the drugs or processes such as drug transport and bacterial metabolism that are known to influence drug interactions (Chevereau & Bollenbach, 2015). DIR genes connected to energy metabolism might be necessary for facilitated transport of the drugs (Allison *et al*, 2011) or may be connected to cellular damage induced by antibiotic-related redox changes (Kohanski *et al*, 2007, 2010; Dwyer *et al*, 2014;

Lobritz *et al*, 2015). Overall, we found that 15% of the *E. coli* genome account for 95% of the predictive ability of our model, and this core set is enriched for genes that are conserved across species.

Our study goes beyond existing drug combination discovery approaches by leveraging chemogenomic data in *E. coli* to infer antibiotic interactions in *S. aureus* and *M. tuberculosis*. Our approach may be especially relevant for *M. tuberculosis*, where combination therapy is critical and experimental screens are difficult, expensive, and time-consuming. Identifying synergistic combinations can help reduce the duration of tuberculosis combination therapy, which currently takes 6 months to 2 years (Ramon-Garcia *et al*, 2011).

INDIGO identified combinations that are synergistic in *E. coli*, but are surprisingly antagonistic in *S. aureus*. The discovery of such combinations that are selectively synergistic (narrow-spectrum synergy) could be used to improve specificity of antibiotic combination therapies to target pathogen strains without significantly affecting beneficial microbes. Strong synergistic combinations in model organisms that are predicted by INDIGO to be conserved in pathogens could be prioritized for testing as "broad-spectrum synergistic combinations". Species-specific interaction outcomes have also been observed for drug combinations against fungal pathogens, similar to our observation in bacteria (Wildenhain *et al*, 2015).

While this capability to predict drug interactions by mapping orthologous genes is a significant advance for this field and can be extended to any organism with genome sequence information, a limitation of this approach is that INDIGO cannot explicitly model the contribution of pathogen DIR genes that are not present in *E. coli*. Our interaction estimates for these systems could be further improved with the availability of chemogenomics data for these pathogens and by performing drug-interaction measurements directly in pathogens as training data for INDIGO. Predictive accuracy of INDIGO can be enhanced in the future through new technologies for the accurate measurement of drug–drug and chemical–genetic interactions, performing chemogenomic screens with essential genes (Cameron & Collins, 2014), and by harnessing drug physicochemical properties (Yilancioglu *et al*, 2014) and chemical structure (Wildenhain *et al*, 2015). INDIGO can complement theoretical models for predicting multidrug interactions from pairwise interactions (Wood *et al*, 2012), and kinetic modeling-based approaches, which are currently restricted to small pathways due to a lack of known kinetic parameters and drug targets (Singh *et al*, 2015). By transforming drug responses and interactions into the genomic space, our study provides a framework for genomics-driven drug combinations discovery.

## Materials and Methods

### Drug-interaction assays

All drug-interaction experiments were conducted using a $4 \times 4$ checkerboard assay, where the concentration of each drug increases linearly in each axis starting with 0 (no drug). *Escherichia coli* MG1655 or *S. aureus* ATCC 29213 were grown in tryptic soy buffer media in the presence of antibiotics. In *E. coli* experiments, optical density (OD) measurements were done every 15 min for 12 h in a TECAN Infinite F200 microplate reader. Growth rate was then

estimated based on the area under the growth curve. Since the *E. coli* area under the growth curve was highly correlated with the end-point readings, only end-point OD measurement after 12 h was used for *S. aureus* to expedite experiments. For each drug-interaction experiment, we ensured that for each individual drug, there were > 50% inhibition at the highest dose used and < 50% inhibition at the lowest dose used as a quality control check. All drugs were purchased from Sigma. MICs for each drug are provided in Table 1 and Appendix Table S2.

### Quantifying drug interactions

Interactions were quantified based on the isophenotypic growth contour method described in Cokol *et al* (2011) based on the Loewe's additivity model (Loewe 1953). The growth contour (line of constant growth) is linear for noninteracting drugs. Depending on the concavity or convexity of the growth contour, interactions are classified as synergistic or antagonistic. Our null hypothesis based on the Loewe's interaction model is that a drug is noninteracting with itself. Deviations from this null model lead to either synergy or antagonism. The advantage of this approach is that there is no underlying assumption that the data should be normally distributed with similar numbers of synergy and antagonism, or that neutral interactions should be more common than synergy or antagonism.

### INDIGO cross-validation and controls

#### Alternate metrics to evaluate the predictive ability

We assessed the overall accuracy of the predictions by measuring the area under the receiver-operating characteristics (ROC) curve. INDIGO quantitatively predicted drug interactions with high precision, significantly better than random, and with equal accuracy for both synergy and antagonism (AUC for synergy = 0.79, *P*-value = $10^{-16}$; AUC for antagonism = 0.8, *P*-value = $10^{-16}$). Both the correlation-based and profile overlap-based approaches from Jansen *et al* were significantly less accurate than INDIGO in predicting drug interactions (AUC for synergy = 0.6; *P*-value = 0.003 for profile correlation, and AUC synergy = 0.64; *P*-value = $10^{-5}$ for profile overlap; Fig 3E and F). In addition to these metrics for evaluating accuracy, we also measured the probabilistic concordance index, PCC (Bansal *et al*, 2014), which quantifies the concordance between the ranking of drug pairs in the predicted and measured data, accounting for variance in the experimental measurement of the drug interactions. INDIGO's predictions were significantly better than predictions based on profile similarity and random permutation of the data (INDIGO PCC = 0.68, *P*-value < $10^{-6}$; profile similarity PCC = 0.53; *P*-value = 0.15). We also tested our model in a single-dose study involving 21 antibiotic pairs (Yeh *et al*, 2006) and found that INDIGO outperformed similarity-based approaches in predicting drug interactions [rank correlation *R* = 0.59, *P*-value < $10^{-15}$ for INDIGO; no significant correlation for correlation and profile overlap (Appendix Fig S15)].

#### Estimating robustness to training data

In addition to the test-set predictions, the model's predictive ability was also assessed through cross-validation. Two types of cross-validation analyses were performed. In tenfold cross-validation, 10% of the interactions were randomly blinded and predicted by the model

based on information from the remaining 90% of the interactions (Appendix Fig S10). This represents a scenario where new interactions are inferred for drugs for which some interactions with other drugs are already known. We also performed another kind of cross-validation where each drug was removed from the network and its interactions were inferred based on its chemogenomic profile and interactions of the remaining drugs. This is a more stringent test than tenfold cross-validation and represents the situation where interactions are inferred for a new drug based on its chemogenomic profile (Appendix Figs S9 and S10). Through these analyses, we found that INDIGO could accurately predict interactions with compounds that belong to novel chemical classes or with distinct mechanisms of action. Nevertheless, we found that the prediction accuracy could be further improved by choosing drugs representative of different classes in the training set. In addition, the presence of a few known interactions about a drug of interest can greatly increase predictive ability. Finally, INDIGO was able to predict synergistic interactions with high accuracy in the test set, which had significantly more synergistic interactions (13 synergies and 25 antagonisms) compared to the training data (14 synergies and 56 antagonisms), suggesting that it is robust to the biased distribution of interactions in the training set.

#### Negative and positive controls for INDIGO

As a negative control, we shuffled the interaction scores for the training data and found that it removed correlation and predictive ability of the model. Swapping drug names, which results in switching the chemogenomic profiles of individual drugs, also led to a significant decrease in correlation compared to a random model, confirming that chemogenomic profiles provide unique drug-specific information (Appendix Figs S11 and S12). To determine the role of interaction network topology in influencing the interaction predictions ("hub effect"), we also compared predictions with a simple model that outputs for a query drug, the average interaction score of the partner drug with other drugs in the network. This model had a Pearson's correlation of only 0.3 based on cross-validation, ruling out the possibility that the predictions were primarily influenced by few synergistic and antagonistic hubs in the drug-interaction network. As a positive control, we found that INDIGO correctly identified interactions between the same drug at different doses to be linear (median score = 0 for self–self interactions involving 73 drugs; Appendix Fig S16). This indicates that INDIGO can differentiate chemogenomic profiles of the same drug at different doses from other drugs. This is significant given that the model was trained only on interactions between different drugs.

### Chemogenomic profile similarity-based approaches

For the correlation method, we measured the Pearson's correlation between the chemogenomic profiles of two drugs of interest. Interactions were defined to be synergistic, antagonistic, or linear if the correlation passed a specific threshold. The best threshold was determined based on the training data using the random forest algorithm. Similarly, profile overlap was defined as the overlap in the number of sensitivity genes between the chemogenomic profiles of two drugs of interest. We used a hypergeometric test to determine whether the total overlap was significantly higher than that

    

expected by random chance. The best *P*-value threshold for differentiating different interaction types was determined based on the training data using the random forest algorithm.

### Orthology mapping to *S. aureus* and *M. tuberculosis*

Orthologous genes in *E. coli* were obtained from OrthologeDB (Whiteside *et al*, 2013). The database uses the reciprocal-best-BLAST hit (RBBH) procedure to generate the initial set of ortholog predictions. Genes are declared orthologs if they are each other's top BLAST hit when each genome is BLASTed against the other (Whiteside *et al*, 2013). This resulted in 844 and 909 genes that were predicted to be orthologs of *S. aureus* and *M. tuberculosis* among the *E. coli* genes in our model. Enrichment for orthologs among the top predictors was done using the hypergeometric test in MATLAB.

### Predicting interactions that differ between *E. coli* and other organisms

To predict interactions that differ between the two species, we identified genes in the *E. coli* model that were not orthologous between the two species. The sigma and delta scores for these genes were then set to be at the minimum value (0). The resulting interaction scores for all the drug-interaction pairs were recalculated using the modified scores. Theoretically, there are millions of possible solutions for changing the sigma score matrix to match *S. aureus* or *M. tuberculosis*. To simplify the problem, we simulate the extreme case and set the states of all non-orthologous genes to be zero. Choosing either the maximal (2) or minimal (0) sigma score value led to the same correlation with the predicted difference ($R = 0.52$). Changing the sigma scores would allow us to identify drug interactions most sensitive to the state of the non-orthologous genes. By sequentially removing interactions that have large predicted deviation scores, we found that we could predict *S. aureus* interactions using INDIGO with a much higher correlation ($R > 0.55$) than experimental *E. coli* data (Appendix Fig S14).

### Algorithm and data availability

The following datasets are provided: (i) experimental drug-interaction scores for all 171 combinations in *E. coli* (Dataset EV1); (ii) interaction predictions for 2628 drug combinations of 73 drugs (highest dose in chemogenomics data was chosen) in all three species (*E. coli*, *S. aureus,* and *M. tuberculosis*) (Dataset EV2), and (iii) ranked list of all genes used by INDIGO for predicting drug-interaction outcomes (Dataset EV3). The MATLAB implementation of INDIGO along with associated experimental data is provided as Computer Code EV1 and is also freely available on Synapse, the online portal for sharing computational biology data and software: https://www.synapse.org/#!Synapse:syn3880435/wiki/.

**Expanded View** for this article is available online.

### Acknowledgements
This work was supported by grants from the Harvard Society of Fellows and the Harvard William Milton Fund to SC, National Institute of Allergy and Infectious Diseases of the National Institutes of Health under Award Number U19AI111276, the Broad Institute Tuberculosis donor group and the Pershing Square Foundation, and the Wyss Institute to JJC, Turkish Academy of Sciences GEBIP Programme to MC. The content is solely the responsibility of the authors and does not necessarily represent the official views of the National Institutes of Health or other funding agencies. We thank Caroline Milne Porter and Michael Lobritz for insightful discussions and critical reading of the manuscript.

### Author contributions
SC conceived, designed, and implemented INDIGO and performed all the computational analyses. MC designed and supervised all the experimental analyses. MC-C, NS, KY, and MC performed experiments; HK reproduced all the computational results and provided feedback; SC, JJC, and MC contributed to the study design; SC wrote the manuscript with input from JJC and MC.

### Conflict of interest
KY and MC are cofounders of SynVera, Inc., a start-up focused on drug combination therapy development. The other authors declare that they have no conflict of interest.

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
