## [Review Process File · Molecular Systems Biology]

Chemogenomics and Orthology-Based Design of Antibiotic Combination Therapies

Sriram Chandrasekaran, Melike Cokol-Cakmak, Nil Sahin, Kaan Yilancioglu, Hilal Kazan, James J. Collins, and Murat Cokol

Corresponding authors: Sriram Chandrasekaran and Murat Cokol, Harvard University

Review timeline:

Submission date:	06 July 2015
Editorial Decision:	26 August 2015
Resubmission :	31 December 2015
Editorial Decision:	11 March 2016
Revision received:	09 April 2016
Editorial Decision:	19 April 2016
Revision received:	20 April 2016
Accepted:	21 April 2016

Editor: Thomas Lemberger

Transaction Report:

1st Editorial Decision

26 August 2015

We have now heard back from the two referees who accepted to evaluate your manuscript. As you will see from the reports below, the referees raise substantial concerns on your work, which, I am afraid to say, preclude its publication.

Both reviewers acknowledge that the ideas behind the presented analysis are good and the general topic of the study interesting. However, the reviewers also feel that the quality of the predictions made remains limited, in particular due to the fact that the only limited new data are provided to demonstrate the broad applicability of the proposed approach. While reviewer #1 is cautiously supportive, reviewer #2 is not convinced that the study would provide sufficient novel insights.

With the rather limited level of support provided by the reviewers, I am afraid I see no choice but to return the manuscript with the message that we cannot publish it.

Nevertheless, we recognize that the subject matter and your approach are of potential interest, and we would not be opposed to consider a new study that extends the present work, provide a more extensive and convincing validation of the presented approach with sufficient novel data. This would have a new number and receipt date. We recognise that this may involve further experimentation and analysis, and we can give no guarantee about its eventual acceptability. However, if you do decide to follow this course then it would be helpful to enclose with your re-submission an account of how the work has been altered in response to the points raised in the present review.

I am sorry that the review of your work did not result in a more favourable outcome on this occasion, but I hope that you will not be discouraged from sending your work to Molecular Systems Biology in the future.

REFEREE COMMENTS

Reviewer #1:

This study presents a novel strategy for rationally designing combinatorial antibiotic treatments with the help of publically available large chemical genomic datasets. The authors build a machine-learning algorithm (using Boolean operations and random forests), INDIGO, which can predict the outcome of combinatorial antibiotic treatments based on the sensitivity profiles of single-gene mutations of *E. coli* to the individual drugs. The model is initially trained and then validated on a drug-drug interaction dataset experimentally obtained by the authors, and slightly improved by integrating a few physicochemical properties of the drugs. The predictions are then extended to the entire collection of drugs and stresses present on the published *E. coli* chemical genomic dataset and the key genes contributing either to antagonism or synergy are extracted as "genetic predictors" of drug-drug interactions. Finally, the authors use orthology to predict the outcome of drug-drug interactions in the Gram+ pathogen, *S. aureus*, using the model initially built on *E. coli* data.

Overall the study provides a solid new framework for predicting drug-drug interactions. Although the idea to use chemical genomics data for predicting drug-drug interactions is not new, and has been tried repeatedly in the past, this is by far the most convincing and thorough effort towards this direction (however see also major comment 5). The implications of a having a workable and transferable pipeline for predicting the effects of combinatorial treatments are immense -especially since this can go beyond the scope of antibiotics/adjuvants and used for drugs against other infectious and non-infectious diseases (e.g. anti-cancer drugs).

Despite how much enthusiastic I am with the ideas, outcome and potential of this work, the manuscript has several flaws- ranging from poor description of analysis pipeline and lack of necessary quality controls to lack of focus/linearity and overstatements. The authors try to deal with way too many aspects, unavoidably leaving holes everywhere (problem formulation, experimental design, method documentation, result presentation and evaluation), breaking the flow of the text and making it harder to follow (e.g. relevant information for a given topic is often scattered between the main text, methods section and figure legends; clear definitions are scarce -e.g. what do authors mean by linear or non-linear interactions?- and in contrast different wording for the same concept -recall=sensitivity).

That being said, I find all limitations addressable -some re-analysis is needed to solidify some of the conclusions and extensive re-writing (including some rethinking of focus) is required for making the story easy-to-follow by both a non-expert and an aficionado. The core results, concepts and message of the paper are certainly worth to be read by a broad audience.

Major comments

1. A good description of INDIGO is missing (Fig 1 and methods fail to do this). No clear description of the model input and model parameters (if any) are present in the entire manuscript. For example, what exactly are (and how many) the features and samples for the random forest input? Because this information is not present, the reader cannot understand what the authors mean with "sigma score", mentioned only at a late stage of the manuscript and never defined. Note that this renders the adaptation for the *S. aureus* model impossible to follow (see also 6th comment)
2. There are a number of points not explained well/ not addressed in the drug-drug interactions tested by authors (either to train or validate their model).

First, I would assume the null hypothesis is that most drugs have neutral interactions. However, the set used to train the model is extremely biased to antagonisms (56 vs 35 neutrals and 14 synergies). Does this imply a biased dataset, a high false positive rate for antagonisms, and/or just much more

room to detect antagonism (one can address antagonism in 8/9 drug concentrations tested) than neutrality (2/9) and synergy (1/9)? If/how that would affect predictions and whether there is room for improvement would be a great point of Discussion (see also comments 3, 9 & 10).

Second, it is very vague how interactions are quantified with visual representation making things worse (background coloring for synergy/antagonism in Figs 2 and 3A makes it impossible to differentiate the strength of growth). Among many examples, here are a few to illustrate the point:
 -why AMK+CLA is qualified synergistic and not additive (Fig 2)?
 -why is the synergy weighed differently for CIP-TRI and TET-CLA (Fig 2)?
 -NAL-FUS is additive and FUS-RIF is strong synergy, although they have the same pattern (Fig. 3A).

Third and most importantly, quality control for the experimentally measured interactions is insufficient:

-what are the axes for replicate reproducibility in Fig 2? I assume some representation of growth (AUC) with a maximum threshold, but it is not stated. Also are all points of the 4x4 matrix for 105 combinations (1680 points) present in this scatter plot?
 -how do authors deal with the large variation of the single-drug responses- e.g. AMK and TRI (but many others too) sometimes kill in the 2nd highest concentration on their own, sometimes in 3rd and sometimes not at all... Is this variance/experimental error taken into account when calling interactions?

Taking into account these limitations, my suggestion would be:

- a) to address replicate reproducibility also on the basis of the α -score (calculate it for each of the replicates); this would be a better metric to get an idea on how many of the interactions are reliable.
- b) benchmark interactions on the Yeh et al. dataset -i.e. how many of the drugs show same interaction? You have 11/15 drugs in common (14/19 in total).

3. H2O2 exhibits strong antagonism with each one of the 18 drugs tested! Do the authors think this is real or an artifact? How much does it affect their model (see 4th comment)?

Assuming that at least some/most of these interactions are real, it would be helpful if authors reconcile this finding with the oxidative stress-bactericidal antibiotics model. Increasing ROS in the cell is reported to enhance bactericidal antibiotic action (Brynildsen, Nat Biotech 2013) -here you see the opposite. Also both from original papers and follow up work from the Collins lab, one would expect bactericidal and bacteriostatic antibiotics interacting differently with H2O2. Although I understand that solving this conundrum is not the purpose of this paper, I still think this unexpected result warrants some discussion.

4. Looking into the experimental validation of the model (Fig. 3), the 10 top predicted antagonisms all involve either H2O2 or a DNA stressing antibiotic. Going back to 1st comment, is it possible that the training set for this model is biased to predict antagonism involving oxidative stress/DNA damage, thus introducing a high rate of false positives on one hand, and on the other hand missing interactions between drugs with other targets (false negatives)? Seeing the precision and ROC curves on antagonisms - Fig S2 and S3, this seems to be an issue.

Would the leave-one-drug-out cross validation analysis presented on Fig S4 help to solve this question? It seems that removing H2O2 from the training set leads to a striking loss of prediction ability (at least of H2O2 interactions), implying that this compound may have a very strong influence on the outcome of the machine-learning algorithm. An easy-to-interpret figure assessing how the loss of each drug influences the prediction of all interactions (both experimental and validation set), not only for the same drug, would help evaluating what drugs help the model the most/least and to detect potential biases.

5. Authors compare their pipeline with Jansen et al. 2009 (based on chemogenomic profile similarity) - but they have missed an important deviation of the same concept introduced by Brown JCS et al. Cell 2014, termed O2M. This is certainly a more advanced way of using the chemogenomic profile similarity to predict synergies (by identifying biomarker genes). Comparisons with the O2M algorithm (ROC curves) should be part of the manuscript.

6. The ability to even partially predict interactions in *S. aureus* with *E. coli* data is remarkable. Yet, the pipeline is poorly described and many of experimental/data analysis aspects are vague:

- a) was there a reason to test only a subset of drugs in *S. aureus*?
- b) why is there a different metric used to assess growth in *S. aureus* (endpoints vs AUC)?
- c) what is the reproducibility of *S. aureus* data? Since noise of single drugs is similar as in *E. coli* data (see 2nd comment)- how is this dealt with?
- d) what are the sigma scores calculated for genes (in Methods)? If you are perturbing the sigma scores for the non-common genes (setting them either to max or min value) then you should have millions of solutions for the score matrix. Or do you set all of them together at min or max value?

In addition a plot of the predicted vs measured interaction score in *S. aureus* would be very useful. If the model is improving predictions, then the correlation should be higher than the one you have for the measured interactions in the 2 organisms (0.48; Fig. 6C).

7. There is a number of data/info that should be made public available in a "raw" (tabulated/txt) format, so reader can evaluate, analyze or re-use data:

- a) drug concentrations and growth data for all drug-dosage matrixes (Fig. 2, 3A, 6B and S1) - at least the growth metric used in this study (AUC)
- b) 324X324 drug interaction matrix -generated by INDIGO using all conditions screened in the *E. coli* chemical genomics work
- c) set of genes that account for 50%, 75% and 95 % of INDIGO predictions

8. There are parts of the paper that are poorly described and/or unnecessary for the main conclusions (breaking its linearity). The authors should frame/explain them better, if they wish to keep them, or remove them:

- a) Reference to many of the Supplementary Figures is poor in main text -e.g. Figs S3-S9 are all introduced in one general sentence. Some of them are very important (e.g. S3-S5), others less- but in any case would need a discussion on what is the message - split between main text and perhaps an additional supplementary text.
- b) Fig S10 explanation in text and legend: too technical and cryptic for reader to understand.
- c) Benchmarking the INDIGO algorithm against the Yeh study data has limited value, since many drugs in Yeh et al. are common with initial drugs used for building the model (on the contrary, it can be used to benchmark the experimental data- 2nd comment)
- d) Role of drug dosage. Here I am missing the point entirely. If it is that drug interactions have to be probed in a dosage-dependent manner, then I would say is nothing new. If it is that INDIGO can predict well interactions calculated either by Bliss or Loewe, then I would say semantics- boils down to the fact that Bliss or Loewe gives you similar solutions. If there is something that would be interesting for audience would be to address how drug dosage affects the predictive ability of the model INDIGO comes up with; the *E. coli* chemical genomics datasets have several concentrations for each and it is unclear if some are more/less helpful in building the model.

9. Some of the interpretations (that are actually peripheral to the main messages of this paper) are not entirely justified by data presented in this paper. They may fit better to Discussion- using more cautious statements:

- a) are antagonistic interactions more often than synergistic (or using Loewe additivity for a 3x3 combination matrix creates biases - see comment 1)? There is better evidence -by the same authors- for antagonisms being more prevalent in antifungals.
- b) are drug interactions driven by drug-target? Aminoglycosides are far away from other protein synthesis inhibitors, AMK is not with rest of aminoglycosides, NIT has DNA damage as main cytotoxic effect, but clusters with aminoglycosides. So best case this statement is to first approximation.
- c) why are the genes important for predicting drug interactions relevant as drug-targets? Is the assumption that genes involved in the drug-drug interactions (Chevereu & Bollenbach, 2015) are the same as genes that are good "genetic predictors" for interactions? If true, it would be nice to provide evidence for this.

10. Discussion would benefit if it were more focused (targeted more to discussing the results rather than potential of method) and structured (subsections). Assessment of model and data used to build model, how each can improve (e.g. data for essential genes in chemogenomics), what would be needed to be expandable to other organisms, how chemogenomics data from different organisms can

be used to assess/improve ability to predict interactions in other organisms are all relevant points for discussion.

Also some caution on the use of buzz-words and the claims would be beneficial - e.g. I would not call INDIGO an evolutionary algorithm (all done was reciprocal blasting to identify orthologues) nor have I seen any evidence that differences in effectiveness of drug combination can be spotted on a strain/"personal medicine" level (i.e. using *E. coli* data you can predict the differences in drug synergies over different clinical isolates of *S. aureus*).

Minor comments

1. Why is only drug sensitivity (and not resistance) used from the chemical genomics data? What is the reason for the renormalization of the chemical genomics data? Data scores are already normalized and represent adjusted t-tests according to authors. Quantile normalization and z-scoring t-tests on top sounds unnecessary/harsh.
2. It is unclear how authors translate α -scores - calculated based on Loewe additivity to a classification that was based on scores calculated based on Bliss independence (Yeh et al., Nat Gen 2006). Yeh et al rescale their "Bliss" scores to -1 to +2; do authors rescale their scores and what is the score range?
3. Rifampicin is not targeting RNA metabolism, but RNA polymerase.
4. The most fundamental limitation of correlation approaches for inferring drug-drug interactions is that they only account for synergies (not antagonisms). Synergies between distinct classes of drugs can be captured (as they may still have similarity in their chemogenomic profiles).
5. Fig. 4: a proper explanation on how the interactions were "collapsed" into the major drug families and target process is absent from the manuscript.
6. The fact that the mean error in prediction correlates with physicochemical properties is convincing, but at the same time intriguing. It would be worth discussing why would a large chemical genomic dataset miss this.
7. Figure 5: Fusidic acid is not the best example; some predictions are getting better with physicochemical data (mentioned by authors), but others are worse.
8. Figure 6- panels are mixed in description in text. Please also define in figure and/or legend that panel C is about measured interaction scores (not predicted). Common genes between *E. coli* and *S. aureus* are the minority, not the majority as Fig 6A implies.
9. Consistency in Figures, Figure panels and corresponding text would be beneficial; for example giving AUC and p-value in all ROC curves in the graph (rather than alternating it in text and legend) would help.
10. In methods there are reported 2 AUC's of 0.68 with very different significances (for model with biophysical properties alone)
11. For many of the Supplementary Figures legends are not enough to follow the figure- this is more of a problem since reference to them in main text is often very minimal.

Reviewer #2:

In this manuscript, Chandrasekaran et al. develop a computational approach aimed at predicting drug interactions between antibiotics from their chemogenomic profiles. The authors developed an algorithm which they trained on a large data set of drug interactions and then used it to predict interactions with antibiotics that were not in the original data set. The authors show that their approach can make successful predictions at a significantly higher rate than two alternative approaches. Their analysis further reveals cellular pathways and physicochemical properties of the

drugs that play a central role for predicting drug interactions. Finally, the authors generalize their approach to predict drug interactions in a different species (*S. aureus*) from chemogenomic data that was obtained in *E. coli* -- a potentially useful approach since large chemogenomic data sets are currently only available for few model organisms.

The study of antibiotic combinations and drug interactions is certainly a timely and relevant topic as evidenced by numerous recent studies that led to considerable conceptual progress on this topic. Computational approaches for predicting drug interactions could lead to important advances in drug combination design since, if successful, they could enable the systematic exploration of the extremely large space of possible drug combinations without a need for cumbersome large scale experimental screens. The present work makes an interesting attempt in this direction and appears technically sound. However, the conceptual advance made in this work remains slightly obscure and the quality of the computational predictions made is not entirely convincing (see major points below). Overall, while the topic is interesting, this manuscript in its current form might be better suited for a more specialized bioinformatics journal.

Major points:

1. A serious concern is that the predictions made by the authors' algorithm are not entirely convincing. Specifically, these predictions require a large training set of 15 drugs and are only tested on a considerably smaller set of 4 additional drugs. Even then, the quality of the predictions is relatively limited, e.g. the scatterplot of the predicted and measured interaction scores in Fig. 3C shows a weak rank correlation but it seems that only the strongest effects are predicted to some extent while the rest looks essentially random. A similar problem is apparent in Fig. 6D where the observations for *S. aureus* are compared to the predictions: the weak correlation that is observed seems to depend entirely on 3 or 4 data points (located in the top right and bottom left). Is this correlation significant? Independent of statistical significance, it looks as if only a small fraction of the differences are correctly predicted. The usefulness of this approach is thus doubtful and needs to be clarified.
2. The "INDIGO" algorithm is central to the results of the study. However, its description in the main text is inadequate, making it hard to assess the conceptual advance achieved in this work. This algorithm is briefly introduced in the first paragraph of the results part and in Figure 1 but the main idea and procedure used to make the predictions did not become clear to me from this. This part will be particularly hard to understand for a broad audience of biologists. The main idea behind this algorithm should be clarified and its description considerably expanded.
3. The overall approach is conceptually similar to (Yilancioglu et al., *J Chem Inf Model*, 2014) from the same last author. While this previous work focused on yeast and physicochemical properties of the drugs to predict drug interactions, the general approach of using available data on drugs and using machine learning or related approaches to predict drug interactions based on a training data set is conceptually similar. The advance of the present study thus appears incremental. It would need to be clarified how the present study presents a major step forward compared to this previous work.
4. There is little novel experimental data in this work, so the predictions made are largely about the past. E.g. the experimental results shown in Fig. 2 were generated in this study but a similar study was published almost 10 years ago (Yeh et al., *Nat Gen*, 2006) and repeated for other organisms since then (as properly acknowledged by the authors). Similarly, the computational analysis relies almost entirely on the published chemical genomics data set from (Nichols et al., *Cell*, 2011). It is of course perfectly fine to reanalyze existing data but the lack of new data will likely limit the appeal of this study to a more specialized community of bioinformaticians interested in drug interactions. In the absence of any mechanistic insight, the claim that drug interactions can be predicted based on chemical genomics data would be more convincing if the authors could validate predictions with a truly novel data set. One possibility could be to investigate combinations of three or more drugs. Recent studies (e.g. Wood et al., *PNAS*, 2012) made progress in predicting such higher order drug combinations and it would be interesting if the authors could at least discuss if their algorithm can make predictions for the effects of higher order drug combinations. It would strengthen this work if the authors could experimentally verify some of these predictions for a few selected combinations of three or more drugs.

Reviewer #1:

This study presents a novel strategy for rationally designing combinatorial antibiotic treatments with the help of publically available large chemical genomic datasets. The authors build a machine-learning algorithm (using Boolean operations and random forests), INDIGO, which can predict the outcome of combinatorial antibiotic treatments based on the sensitivity profiles of single-gene mutations of E. coli to the individual drugs. The model is initially trained and then validated on a drug-drug interaction dataset experimentally obtained by the authors, and slightly improved by integrating a few physicochemical properties of the drugs. The predictions are then extended to the entire collection of drugs and stresses present on the published E. coli chemical genomic dataset and the key genes contributing either to antagonism or synergy are extracted as "genetic predictors" of drug-drug interactions. Finally, the authors use orthology to predict the outcome of drug-drug interactions in the Gram+ pathogen, S. aureus, using the model initially built on E. coli data. Overall the study provides a solid new framework for predicting drug-drug interactions. Although the idea to use chemical genomics data for predicting drug-drug interactions is not new, and has been tried repeatedly in the past, this is by far the most convincing and thorough effort towards this direction (however see also major comment 5). The implications of a having a workable and transferable pipeline for predicting the effects of combinatorial treatments are immense -especially since this can go beyond the scope of antibiotics/adjuvants and used for drugs against other infectious and non-infectious diseases (e.g. anti-cancer drugs). Despite how much enthusiastic I am with the ideas, outcome and potential of this work, the manuscript has several flaws- ranging from poor description of analysis pipeline and lack of necessary quality controls to lack of focus/linearity and overstatements. The authors try to deal with way too many aspects, unavoidably leaving holes everywhere (problem formulation, experimental design, method documentation, result presentation and evaluation), breaking the flow of the text and making it harder to follow (e.g. relevant information for a given topic is often scattered between the main text, methods section and figure legends; clear definitions are scarce -e.g. what do authors mean by linear or non-linear interactions?- and in contrast different wording for the same concept - recall=sensitivity). That being said, I find all limitations addressable -some re-analysis is needed to solidify some of the conclusions and extensive re-writing (including some rethinking of focus) is required for making the story easy-to-follow by both a non-expert and an aficionado. The core results, concepts and message of the paper are certainly worth to be read by a broad audience.

We thank the reviewer for the feedback. We have significantly rewritten the manuscript to address the reviewer's comments and suggestions. To streamline the main message of the text, we have moved specific sections to the supplement as suggested by this reviewer. We have expanded the description in each section so that each section stands independently on its own without referring to different parts of the manuscript.

Major comments

1. A good description of INDIGO is missing (Fig 1 and methods fail to do this). No clear description of the model input and model parameters (if any) are present in the entire manuscript. For example, what exactly are (and how many) the features and samples for the random forest input? Because this information is not present, the reader cannot understand what the authors mean with "sigma score", mentioned only at a late stage of the manuscript and never defined. Note that this renders the adaptation for the S. aureus model impossible to follow (see also 6th comment)

We have considerably updated the section describing INDIGO to clarify the inputs and outputs (described below).

The input to INDIGO consists of: (i) chemogenomic profiles of individual drugs of interest, and (ii) interaction scores for a pair of drugs or chemical agents. INDIGO infers genes that increase sensitivity to the drugs of interest from the chemogenomic profiles (z-score < -2). INDIGO then integrates the profiles of a combination of drugs using Boolean set operations that capture similarity (sigma) and dissimilarity (delta score). In this framework, the union (sigma score) and intersection (delta score) operations are proxies of molecular redundancy or similarity and uniqueness of mechanism of action of the individual drugs (Figure 2). This joint profile, which captures similarity and dissimilarity between two drugs' profiles, is then used as input to a random forest regression algorithm.

2. There are a number of points not explained well/ not addressed in the drug-drug interactions tested by authors (either to train or validate their model). First, I would assume the null hypothesis is that most drugs have neutral interactions. However, the set used to train the model is extremely biased to antagonisms (56 vs 35 neutrals and 14 synergies). Does this imply a biased dataset, a high false positive rate for antagonisms, and/or just much more room to detect antagonism (one can address antagonism in 8/9 drug concentrations tested) than neutrality (2/9) and synergy (1/9)? If/how that would affect predictions and whether there is room for improvement would be a great point of Discussion (see also comments 3, 9 & 10).

We thank the reviewer for this suggestion. This preponderance of antagonism over synergy has been observed in other sensitive drug interaction screens (Ocampo et al, 2014; Yeh et al, 2006). So it is possible that it is not a bias in our data set but a natural phenomenon. In the revised version, we highlighted this observation.

The validation set had significantly more synergistic interactions (13 synergies to 21 antagonisms), while the training data had only 14 synergies for 56 antagonisms. Yet INDIGO was able to predict these synergistic interactions with high accuracy. This highlights the fact that INDIGO not only takes in the distribution of interactions, but also the genetic markers in the chemogenomic profiles of the underlying drugs that lead to strong synergy or antagonism. Hence it is less prone to biases in the training data.

Despite the bias in the training data, INDIGO accurately predicted self-self interactions of the same drug at different doses (represented by different chemogenomic profiles) to be linear or non-interacting. Hence INDIGO is not affected by the bias or preponderance of antagonism in the training set. We found this to be true for 951 interactions between the same 73 drugs (Supplementary Figure 16).

Second, it is very vague how interactions are quantified with visual representation making things worse (background coloring for synergy/antagonism in Figs 2 and 3A makes it impossible to differentiate the strength of growth). Among many examples, here are a few to illustrate the point: - why AMK+CLA is qualified synergistic and not additive (Fig 2)? -why is the synergy weighed differently for CIP-TRI and TET-CLA (Fig 2)? -NAL-FUS is additive and FUS-RIF is strong synergy, although they have the same pattern (Fig. 3A).

We have now displayed the growth data for both the replicates separately in the supplement. For each replicate drug interaction experiment, one alpha score is produced, which is 0 if the drug pair is additive, negative if synergistic, and positive if antagonistic. For each drug pair, the average of two replicates was defined as alpha. Some interactions, such as AMK + CLA, are strongly synergistic in one replicate and weakly synergistic in the other. By looking at both the replicates, it is clear why the interactions are quantified as synergistic. Similarly, NAL-FUS and FUS-RIF are also different in individual replicates. In the revised paper, we describe the drug interaction experiment setup and scoring in greater detail.

Third and most importantly, quality control for the experimentally measured interactions is insufficient: -what are the axes for replicate reproducibility in Fig 2? I assume some representation of growth (AUC) with a maximum threshold, but it is not stated. Also are all points of the 4x4 matrix for 105 combinations (1680 points) present in this scatter plot?

The scatter plot in the original version of Figure 2 has the alpha scores for each replicate of the 105 interactions. The correlation between the interaction scores for both replicates is very strong ($R = 0.81$, p -value = 10^{-26}), indicating high data quality. In the revised paper, an analogous scatter plot is shown in Figure 1.

-how do authors deal with the large variation of the single-drug responses- e.g. AMK and TRI (but many others too) sometimes kill in the 2nd highest concentration on their own, sometimes in 3rd and sometimes not at all... Is this variance/experimental error taken into account when calling interactions?

Each drug interaction experiment conducted during this study was required to pass two quality tests to be included in the final data set: (i) for each individual drug, there should be >50% growth inhibition at the highest dose used, and (ii) for each individual drug, there should be <50% growth

inhibition at the lowest dose used. After a drug interaction experiment passes these quality tests, the largest isophenotypic contour is found and used to compute an interaction score (α). Hence, this variance between single-drug responses is taken into account for drug interaction scoring.

Taking into account these limitations, my suggestion would be: a) to address replicate reproducibility also on the basis of the α -score (calculate it for each of the replicates); this would be a better metric to get an idea on how many of the interactions are reliable.

In addition to the replicates of alpha scores we had earlier in Figure 2, we have also estimated replicate correlations of AUC values. Both rank correlation and spearman correlation show strong correlation between replicates ($R = 0.77$).

In addition, while evaluating interactions, we have used the Probabilistic Concordance Index, which takes into account the variance between replicates to evaluate the predictive ability of the model.

b) benchmark interactions on the Yeh et al. dataset -i.e. how many of the drugs show same interaction? You have 11/15 drugs in common (14/19 in total).

We thank the reviewer for this suggestion. In the revised paper, we benchmark our experimental data with Yeh et al. Twelve of the 19 drugs we used in this study overlap with Yeh et al., which corresponds to 66 interactions. Among the 66 interactions, only one disagreed completely, i.e., it was synergistic in one data set and antagonistic in the other. Twenty seven were exact matches, while the remaining 38 were predicted to be synergistic or antagonistic in one data set, but predicted as additive in the other. The agreement between the two data sets is striking, especially given the fact that: (i) Yeh et al. used a 2x2 dose combination matrix while we used a more sensitive 4x4 matrix, and (ii) Yeh et al. used LB as their growth media, while we used TSB. We have updated the corresponding methods section in the manuscript as follows:

Comparison with existing drug interaction data: We chose drugs that overlap with Yeh et al study to compare our approach to existing data sets in literature. While we have done a multi-dose study, existing studies such as Yeh et al are single-dose measurements. Among the total 1539 dose-specific measurements (171 pairs x 9 dose combinations) in our study, only 66 dose-specific measurements overlaps with Yeh et al. By using the bliss metric to evaluate our interactions and matching the dose used by Yeh et al, we found that the overall correlation between the two data sets among the 66 interactions that were shared was 0.42 (p-value = 0.0004). In addition to dose, the differences in growth media and the assay used for assessing growth, also affects the correlation between the two data sets. Among the 66 interactions, only one interaction disagreed completely i.e was synergistic in one data set and antagonistic in the other. 27 were exact matches, while the remaining 38 were predicted to be synergistic or antagonistic in one data set but predicted as non-interacting in the other.

3. H202 exhibits strong antagonism with each one of the 18 drugs tested! Do the authors think this is real or an artifact? How much does it affect their model (see 4th comment)? Assuming that at least some/most of these interactions are real, it would be helpful if authors reconcile this finding with the oxidative stress-bactericidal antibiotics model. Increasing ROS in the cell is reported to enhance bactericidal antibiotic action (Brynildsen, Nat Biotech 2013) -here you see the opposite. Also both from original papers and follow up work from the Collins lab, one would expect bactericidal and bacteriostatic antibiotics interacting differently with H202. Although I understand that solving this conundrum is not the purpose of this paper, I still think this unexpected result warrants some discussion.

We agree that this is a surprising observation and needs further analysis. We would like to first mention that the dosage of hydrogen peroxide used in this study is 50x higher than used in Brynildsen et al. The treatment times are also much longer (days in our study vs 2 hr in Brynildsen et al). At this high concentration, hydrogen peroxide has been observed to be bacteriostatic (Imlay, 2015) and hence leads to antagonism with many antibiotics. This is consistent with other studies from Collins lab and other groups (Lobritz et al. and Ocampo et al.). Hence our results are not directly comparable with Brynildsen et al., but they are consistent with other studies. We have updated our manuscript to address this difference with the Brynildsen et al. study as follows: We also observed very strong antagonism of hydrogen peroxide with other antibiotics, not observed in a previous study (Brynildsen et al., 2013). This could be because the dosage of hydrogen peroxide used in this study was 50-fold higher, and the treatment times were much longer (24 hours in our

study vs 2 hours) compared to Brynildsen et al. (Brynildsen et al., 2013). At this high concentration, hydrogen peroxide has been observed to be bacteriostatic (Imlay, 2015) and hence leads to antagonism with many antibiotics, which is consistent with other studies (Lobritz et al., 2015; Ocampo et al., 2014).

4. Looking into the experimental validation of the model (Fig. 3), the 10 top predicted antagonisms all involve either H₂O₂ or a DNA stressing antibiotic. Going back to 1st comment, is it possible that the training set for this model is biased to predict antagonism involving oxidative stress/DNA damage, thus introducing a high rate of false positives on one hand, and on the other hand missing interactions between drugs with other targets (false negatives)? Seeing the precision and ROC curves on antagonisms - Fig S2 and S3, this seems to be an issue.

We thank the reviewer for this insightful comment and we have addressed this problem through several analyses. First, we have repeated the analysis without hydrogen peroxide and its removal does not affect the predictive accuracy based on cross validation (described in detail for the next question). Secondly, despite the abundance of antagonistic interactions in the training data (14 synergies and 56 antagonisms), the model was able to accurately predict synergistic interactions in the test set, where the distribution was less skewed (13 synergies and 25 antagonisms). Third, since the predictions are quantitative, we can still rank interactions and identify those interactions that are relatively more synergistic or antagonistic. Hence even if the interactions are biased towards antagonism, we can reduce false positives by choosing a higher threshold for antagonism. Overall, the broad range of antagonism leads to the lowered predictive ability (measured by AUC) compared to synergy. We have updated the results and discussion section to clarify the effect of antagonism and hydrogen peroxide on INDIGO's predictive ability.

Would the leave-one-drug-out cross validation analysis presented on Fig S4 help to solve this question? It seems that removing H₂O₂ from the training set leads to a striking loss of prediction ability (at least of H₂O₂ interactions), implying that this compound may have a very strong influence on the outcome of the machine-learning algorithm. An easy-to-interpret figure assessing how the loss of each drug influences the prediction of all interactions (both experimental and validation set), not only for the same drug, would help evaluating what drugs help the model the most/least and to detect potential biases.

We have repeated the analysis without hydrogen peroxide and its removal does not affect the predictive accuracy based on cross validation ($R = 0.55$ vs $R = 0.57$ with peroxide; Supplementary Figure 7). Additionally, we still find strong AUCs for synergy and antagonism. Hence despite its promiscuous antagonism, it did not have a strong effect on the algorithm's predictive accuracy. As the reviewer correctly points out, removing H₂O₂ from the training set leads to a striking loss in accuracy for predicting H₂O₂ interactions, but not for other drugs, suggesting that its mechanism of interaction might be radically different from what is reflected in its chemogenomic profile. We have now added a new supplementary figure that displays the results of this analysis without H₂O₂ (Supplementary Figure 7).

5. Authors compare their pipeline with Jansen et al. 2009 (based on chemogenomic profile similarity) - but they have missed an important deviation of the same concept introduced by Brown JCS et al. Cell 2014, termed O2M. This is certainly a more advanced way of using the chemogenomic profile similarity to predict synergies (by identifying biomarker genes). Comparisons with the O2M algorithm (ROC curves) should be part of the manuscript.

In addition to the correlation and overlap-based approaches, we have now compared INDIGO with O2M as well. We show that INDIGO significantly outperforms O2M in predicting synergistic interactions in *E. coli*. We have added a supplementary figure comparing O2M with INDIGO (Supplementary Figure 4). While O2M outperforms Jensen et al., INDIGO's accuracy was superior to that of O2M based on both cross validation and test set validation. Analogous to other similarity-based approaches, O2M also does not have a model for antagonism. We have updated the main text based on this analysis as follows:

INDIGO also significantly outperformed the O2M algorithm (Brown et al, 2014), which is an extension of the Jansen et al overlap-based approach (Supplementary Figure 4). Based on our analysis, we find that similarity and overlap-based approaches fail to correctly predict interaction outcomes for new classes of drugs and lack a model for antagonism.

6. The ability to even partially predict interactions in *S. aureus* with *E. coli* data is remarkable. Yet, the pipeline is poorly described and many of experimental/data analysis aspects are vague: a) was there any reason to test only a subset of drugs in *S. aureus*?

The drugs were chosen to be a subset of the 19 drugs measured in *E. coli*; these included drugs from different classes that were predicted by INDIGO to have the greatest range of variation in their interactions (Supplementary Table 3).

b) why is there a different metric used to assess growth in *S. aureus* (endpoints vs AUC)?

From our *E. coli* data, we found that end point and AUC were highly correlated in *E. coli*. Since end point readings are faster and easier to measure (two factors that are especially important when handling potentially pathogenic bacteria) we used the end-point measurements for *S. aureus*. We have updated the orthology section to discuss the choice of drugs and approach as follows:

We predicted interaction and deviation scores for all 171 drug combinations that were experimentally tested in *E. coli* using this orthology framework. To experimentally validate the predictions in *S. aureus*, 10 drugs from the *E. coli* data set were chosen; these drugs belong to different classes that were predicted by INDIGO to have the greatest range of variation in their interactions between *E. coli* and *S. aureus* (Methods; Figure 5, Supplementary Figure 13, and Supplementary Table 2, 3). The interaction score for all 45 pairs of these antibiotics were then experimentally measured in *S. aureus* using the same setup as *E. coli* (Figure 5b).

And in the methods section:

Since the *E. coli* area under the growth curve was highly correlated with the end point readings, only end-point OD measurement after 12 hours was used for *S. aureus* to expedite experiments.

c) what is the reproducibility of *S. aureus* data? Since noise of single drugs is similar as in *E. coli* data (see 2nd comment)- how is this dealt with?

The correlation between AUC growth values between replicates was 0.73, and the correlation between alpha scores was 0.56 (p-value = 10⁻⁵). We have updated Supplementary Figure 13 displaying staph growth data to include these statistics.

To account for the noise in the measurement, we also used the Probabilistic Concordance Index (PCI) to evaluate the predictions by INDIGO. PCI takes into account the variability between replicates to assess the predictive ability of the model. We found that the results were significant based on PCI (p-value = 0.009) and rank correlation (r = 0.52, p-value = 10⁻⁴) for INDIGO.

d) what are the sigma scores calculated for genes (in Methods)? If you are perturbing the sigma scores for the non-common genes (setting them either to max or min value) then you should have millions of solutions for the score matrix. Or do you set all of them together at min or max value?

We agree with the reviewer that there are millions of possible solutions for simulating *S. aureus* data using the sigma score matrix. To simplify the problem, we simulate the extreme case and set them all to be at the minimal value at the same time. This process corresponds to deleting the contribution of these genes from the *E. coli* INDIGO drug interaction model. Choosing min or max leads to the same set of predictions for conserved and divergent interactions. We have updated the methods section and the orthologous interactions section of the paper to better clarify this point.

To quantify the relative contribution of non-orthologous genes on each drug interaction outcome, the chemogenomic scores of non-orthologous genes in the *E. coli* INDIGO model were set to take minimal value (Methods). This process corresponds to deleting these genes from the *E. coli* INDIGO model. The predictions from this modified model was then compared with the original *E. coli* model predictions. The difference in scores for each interaction, called the deviation score, allowed us to identify drug interactions most sensitive to the state of the non-orthologous genes and therefore are less likely to be conserved between *E. coli* and *S. aureus*.

Methods section:

Theoretically, there are millions of possible solutions for changing the sigma score matrix to match *S. aureus* or *M. tuberculosis*. To simplify the problem, we simulate the extreme case and set the states of all non-orthologous genes to be zero. Choosing either the maximal (2) or minimal (0) sigma score value both led to the same correlation with the predicted difference ($R = 0.52$).

In addition a plot of the predicted vs measured interaction score in S. aureus would be very useful. If the model is improving predictions, then the correlation should be higher than the one you have for the measured interactions in the 2 organisms (0.48; Fig. 6C).

We have added a plot of predicted vs measured interaction score in *S. aureus* showing higher correlation by integrating INDIGO predictions (Figure 5). We find that this correlation ($R = 0.47$) is higher than the measured correlation between *E. coli* and *S. aureus* interaction scores ($R = 0.39$). We have updated the orthology section of the paper to include this information.

- 7. There is a number of data/info that should be made public available in a "raw" (tabulated/txt) format, so reader can evaluate, analyze or re-use data:*
- a) drug concentrations and growth data for all drug-dosage matrixes (Fig. 2, 3A, 6B and S1) - at least the growth metric used in this study (AUC)*
 - b) 324X324 drug interaction matrix -generated by INDIGO using all conditions screened in the E. coli chemical genomics work*
 - c) set of genes that account for 50%, 75% and 95 % of INDIGO predictions*

We have made all these data available as supplementary material or in the companion website on synapse.

- 8. There are parts of the paper that are poorly described and/or unnecessary for the main concussions (breaking its linearity). The authors should frame/explain them better, if they wish to keep them, or remove them: a) Reference to many of the Supplementary Figures is poor in main text -e.g. Figs S3-S9 are all introduced in one general sentence. Some of them are very important (e.g. S3-S5), others less- but in any case would need a discussion on what is the message - split between main text and perhaps an addition supplementary text. b) Fig S10 explanation in text and legend: too technical and cryptic for reader to understand.*

We thank the reviewer for this suggestion. We have now added additional text in the methods section, in a subsection called 'INDIGO: Cross Validation and Controls', that describes in detail all the control and supplementary analysis performed to benchmark our approach. We have also corrected and updated the main text references to the supplementary figures. We have also now removed the figure S10 from the supplement as it is unnecessary for the main conclusions and breaks the flow of the manuscript.

- c) Benchmarking the INDIGO algorithm against the Yeh study data has limited value, since many drugs in Yeh et al. are common with initial drugs used for building the model (on the contrary, it can be used to benchmark the experimental data- 2nd comment)*

Even though many drugs overlapped between the data sets, in terms of total interactions only 66 interactions (38%) did. By using the bliss metric and approximately matching the dose used by Yeh et al., we found that the overall correlation between the data sets was 0.42. In addition to dose, the differences in growth media can also affect the correlation between the data sets. Given that only a minority of interactions overlapped and the dosage and growth conditions used were different between the studies, the fact that INDIGO performed well in this data set is still significant. Overall, we agree with the reviewer that this breaks the flow from the main conclusion of the paper, and we have moved this analysis to the methods section with discussion on cross validation and additional controls.

- d) Role of drug dosage. Here I am missing the point entirely. If it is that drug interactions have to be probed in a dosage-dependent manner, then I would say is nothing new. If is that INDIGO can predict well interactions calculated either by Bliss or Loewe, then I would say semantics- boils down to the fact that Bliss or Loewe gives you similar solutions. If there is something that would be*

interesting for audience would be to address how drug dosage affects the predictive ability of the model INDIGO comes up with; the E. coli chemical genomics datasets have several concentrations for each and it is unclear if some are more/less helpful in building the model.

We agree with the reviewer that the observation that drug dosage affects interactions is not surprising, and this section breaks the flow from the main conclusion of the paper; hence we have removed this section from the main text of the revised paper. This greatly improved the flow of the manuscript since now we use the additive model of drug synergy (Loewe) throughout the manuscript except when benchmarking against Yeh et al.

Similar to the effect of dose on drug interaction data, the dose used in chemogenomics data also influences drug interaction predictions. In general, we matched the closest dose in chemogenomics data to the experimentally used dosage. We found that choosing a dose in chemogenomics data that is significantly different from the dose range in drug interaction data reduced the accuracy in test set and cross validation but not significantly ($R = 0.42$ for test set from $R = 0.52$; Supplementary Table 6; for those drugs for which multiple dose existed). So the predictive ability of the model can be improved by choosing the dose for chemogenomics data that are most relevant for analysis, or by determining the interaction outcomes at multiple doses. We have updated the methods section to highlight the role of dosage in both interaction data and chemogenomics data.

9. Some of the interpretations (that are actually peripheral to the main messages of this paper) are not entirely justified by data presented in this paper. They may fit better to Discussion- using more cautious statements: a) are antagonistic interactions more often than synergistic (or using Loewe additivity for a 3x3 combination matrix creates biases - see comment 1)? There is better evidence - by the same authors- for antagonisms being more prevalent in antifungals.

We do not think the large number of antagonisms is a bias of the methodology as previous studies have also observed more antagonistic interactions than synergy studies (Cokol et al, 2011; Ocampo et al, 2014). This suggests that antagonism is more common in nature than synergy. We have updated the revised version to comment on the higher frequency of antagonism over synergy as follows:

In agreement with previous studies in bacteria and yeast, antagonistic interactions are more prevalent than synergistic interactions. This underscores the difficulty of selecting drug combinations that do not adversely affect clinical outcomes.

b) are drug interactions driven by drug-target? Aminoglycosides are far away from other protein synthesis inhibitors, AMK is not with rest of aminoglycosides, NIT has DNA damage as main cytotoxic effect, but clusters with aminoglycosides. So best case this statement is to first approximation.

We agree with the reviewer that the clustering only approximately matches the drug target processes. There are additional factors that seem to play a role, such as whether the drug is bacteriostatic or bactericidal. We have now rewritten the text to reflect this:

Further, unsupervised clustering of interactions grouped the drugs based on both their mechanism of action, and their bacteriostatic and bactericidal properties, consistent with previous studies (Ocampo et al, 2014; Yeh et al, 2006).

c) why are the genes important for predicting drug interactions relevant as drug-targets? Is the assumption that genes involved in the drug-drug interactions (Chevereu & Bollenbach, 2015) are the same as genes that are good "genetic predictors" for interactions? If true, it would be nice to provide evidence for this.

We found that many genes that are predictors are either involved in the target processes of the drugs or found to influence drug interactions as discovered by Chevereu & Bollenbach (2015). We have updated the relevant discussion to clarify this point:

Genes identified to be top predictors of drug interactions from chemogenomic profiles by INDIGO (DIR genes) were involved in the processes targeted by the drugs or processes such as drug transport and bacterial metabolism that are known to influence drug interactions (Chevereu & Bollenbach, 2015). DIR genes connected to energy metabolism might be necessary for facilitated transport of the

drugs (Allison et al, 2011) or may be connected to cellular damage induced by antibiotic-related redox changes (Dwyer et al, 2014; Kohanski et al, 2010; Kohanski et al, 2007; Lobritz et al, 2015). The top predictors and the associated cellular pathways can be targeted for enhancing synergy between antibiotics.

10. Discussion would benefit if it were more focused (targeted more to discussing the results rather than potential of method) and structured (subsections). Assessment of model and data used to build model, how each can improve (e.g. data for essential genes in chemogenomics), what would be needed to be expandable to other organisms, how chemogenomics data from different organisms can be used to assess/improve ability to predict interactions in other organisms are all relevant points for discussion.

We thank the reviewer for the suggestion. We have toned down the discussion and rewritten this section as suggested by the reviewer.

Also some caution on the use of buzz-words and the claims would be beneficial - e.g. I would not call *INDIGO* an evolutionary algorithm (all done was reciprocal blasting to identify orthologues) nor have I seen any evidence that differences in effectiveness of drug combination can be spotted on a strain/"personal medicine" level (i.e. using *E. coli* data you can predict the differences in drug synergies over different clinical isolates of *S. aureus*).

We agree with the reviewer and have rewritten the section accordingly. We have toned down our remarks on the potential future applications of our approach in the discussion section.

Minor comments

1. Why is only drug sensitivity (and not resistance) used from the chemical genomics data?

As a result of the Nichols et al. study design, there were considerably fewer statistically significant associations for sensitivity than resistance. Nichols et al. report that "80% of the phenotypes were negative (gene deletion more sensitive) and 20% positive (gene deletion more resistant), consistent with recent genetic interaction analyses in *S. cerevisiae* (Fiedler et al., 2009) and *S. pombe* (Roguev et al., 2008)". Hence we decided to use gene sensitivity profiles as they were more abundant and statistically significant interactions in Nichols et al. It should be possible to include resistant genes in the *INDIGO* framework. We have updated the methods section of the paper to clarify this point.

What is the reason for the renormalization of the chemical genomics data? Data scores are already normalized and represent adjusted t-tests according to authors. Quantile normalization and z-scoring t-tests on top sounds unnecessary/harsh.

We did not perform a z-normalization on top of their data set; we used the z-scores from the Nichols study. We performed quantile normalization to ensure that the distributions across drugs were similar to have a uniform interpretation of the z scores. We have rewritten the relevant section of the paper to clarify this.

2. It is unclear how authors translate α -scores - calculated based on Loewe additivity to a classification that was based on scores calculated based on Bliss independence (Yeh et al., Nat Gen 2006). Yeh et al rescale their "Bliss" scores to -1 to +2; do authors rescale their scores and what is the score range?

We did not rescale our alpha scores. Our analyses were performed on quantitative data and predictions by *INDIGO* are quantitative as well. The data were classified for visualization and for estimating AUC values. For such classifications, we had set -0.5 as the threshold for strong synergy and 1 as the threshold for antagonism. We performed sensitivity analysis to show that the model is robust to the choice of values for synergy and antagonism (Supp. Figure 5).

3. Rifampicin is not targeting RNA metabolism, but RNA polymerase.

We agree with the reviewer and have revised the text accordingly.

4. The most fundamental limitation of correlation approaches for inferring drug-drug interactions is

that they only account for synergies (not antagonisms). Synergies between distinct classes of drugs can be captured (as they may still have similarity in their chemogenomic profiles).

We agree with the reviewer and have rewritten the section accordingly as follows:
Based on our analysis, we find that similarity and overlap-based approaches fail to correctly predict interaction outcomes for new classes of drugs and lack a model for antagonism. Thus, taking into account the identity of the individual genes in the chemogenomic profile and accounting for both similarity and dissimilarity, increases the ability to predict drug interactions. Our data shows that drugs with similar targets and chemogenomic profiles can have both synergistic and antagonistic outcomes.

5. Fig. 4: a proper explanation on how the interactions were "collapsed" into the major drug families and target process is absent from the manuscript.

To determine average synergistic or antagonistic interactions between different groups, we compared the distribution of interaction scores for all drug combinations between two groups with the background interaction score for all drug pairs using a t-test. We have updated the figure legend to better describe this as follows:

Figure 5: Global analysis of drug-drug and drug-stress interactions. Panels A and B summarize the interaction outcome between drugs from similar or different families and target processes. The heat-maps represents the entire drug interaction matrix, inferred from the entire compendium of Nichols et al., collapsed into 13 major drug families (panel A) or 10 major target processes (panel B). To determine average synergistic or antagonistic interactions between different groups, we compared the distribution of interaction scores for all drug combinations between two groups with the background interaction score for all drug pairs (Methods).

6. The fact that the mean error in prediction correlates with physicochemical properties is convincing, but at the same time intriguing. It would be worth discussing why would a large chemical genomic dataset miss this.

The connection between physicochemical properties and synergy/antagonism has been observed previously by other studies (Chongsiriwatana et al, 2011; Yilancioglu et al, 2014). However, previous chemogenomics studies have not attempted to predict drug interactions from chemogenomics data. In the revised paper, we remove the analysis on physicochemical properties in order to focus on the main findings on drug interaction prediction and orthology.

7. Figure 5: Fusidic acid is not the best example; some predictions are getting better with physicochemical data (mentioned by authors), but others are worse.

We agree with the reviewer that this section breaks the flow from the main conclusion of the paper (referring to major comment #8); hence we have removed this figure and the corresponding section from the manuscript.

8. Figure 6- panels are mixed in description in text. Please also define in figure and/or legend that panel C is about measured interaction scores (not predicted).

We have updated the figure legend of panel C to emphasize that the interactions are experimentally measured and not predicted by INDIGO.

Common genes between E. coli and S. aureus are the minority, not the majority as Fig 6A implies.

Common genes between the two species are enriched in the top predictive gene set. But overall, we agree with the reviewer that common genes are the minority and have updated the figure panel accordingly.

9. Consistency in Figures, Figure panels and corresponding text would be beneficial; for example giving AUC and p-value in all ROC curves in the graph (rather than alternating it in text and legend) would help.

We have updated the figures and figure legends so that the AUC scores and the p-values are mentioned in the figure legend.

10. In methods there are reported 2 AUC's of 0.68 with very different significances (for model with biophysical properties alone)

The p-values were obtained from different distributions. It is harder to predict synergy than antagonism; so an AUC value of 0.68 for synergy is statistically more significant than an AUC value of 0.68 for antagonism. We have updated the relevant section (Supplementary Figure 6) to highlight the reason why the p-values were different.

11. For many of the Supplementary Figures legends are not enough to follow the figure- this is more of a problem since reference to them in main text is often very minimal.

We have rewritten many of the supplementary figure legends so that they can be assessed independently without referring to the text in the methods section.

Reviewer #2:

*In this manuscript, Chandrasekaran et al. develop a computational approach aimed at predicting drug interactions between antibiotics from their chemogenomic profiles. The authors developed an algorithm which they trained on a large data set of drug interactions and then used it to predict interactions with antibiotics that were not in the original data set. The authors show that their approach can make successful predictions at a significantly higher rate than two alternative approaches. Their analysis further reveals cellular pathways and physicochemical properties of the drugs that play a central role for predicting drug interactions. Finally, the authors generalize their approach to predict drug interactions in a different species (*S. aureus*) from chemogenomic data that was obtained in *E. coli* -- a potentially useful approach since large chemogenomic data sets are currently only available for few model organisms.*

The study of antibiotic combinations and drug interactions is certainly a timely and relevant topic as evidenced by numerous recent studies that led to considerable conceptual progress on this topic. Computational approaches for predicting drug interactions could lead to important advances in drug combination design since, if successful, they could enable the systematic exploration of the extremely large space of possible drug combinations without a need for cumbersome large scale experimental screens. The present work makes an interesting attempt in this direction and appears technically sound. However, the conceptual advance made in this work remains slightly obscure and the quality of the computational predictions made is not entirely convincing (see major points below). Overall, while the topic is interesting, this manuscript in its current form might be better suited for a more specialized bioinformatics journal.

Major points: 1.

A serious concern is that the predictions made by the authors' algorithm are not entirely convincing. Specifically, these predictions require a large training set of 15 drugs and are only tested on a considerably smaller set of 4 additional drugs.

We understand the reviewer's concern. We would like to point out that our analysis is at the interaction level. For every addition of a drug, the number of interactions increases exponentially. The addition of four drugs is a 70% increase in total interactions measured. The training was done on 91 interactions and experimentally validated on 66. Hence the validation data set is not considerably small, but 72% of the size of the training set. Further to training on the experimental data set, we have applied our approach to predict interactions for over 52,000 dose-specific combinations. The scale of interactions that can be predicted by our approach is hence orders of magnitude larger. Most importantly, the method, which we demonstrate for *S. aureus* and *M. tuberculosis*, can be extended to other organisms. Therefore, we think this is a significant advance for this field. We would also like to bring to the reviewer's attention that we experimentally tested our predictions for 45 combinations in *S. aureus* (all pairwise interactions of 10 antibiotics), all of

which are novel.

Even then, the quality of the predictions is relatively limited, e.g. the scatterplot of the predicted and measured interaction scores in Fig. 3C shows a weak rank correlation but it seems that only the strongest effects are predicted to some extent while the rest looks essentially random.

As the reviewer correctly points out, our model predicts the strongest effects, i.e., the strongest synergies and antagonisms, which we believe are potentially of more interest than other weaker interactions. Nevertheless, we have assessed the overall statistical significance of our approach through several metrics. Both parametric and non-parametric approaches show that the correlation between model predictions and experimental observation is significant. We also used the probabilistic concordance index (PCC), which takes into account the noise in the data and the results were still significantly better than random (p-value < 10⁻⁶). We found that even after removing the strongest synergistic or antagonistic interactions, the correlation is still significant (r = 0.46, p-value = 10⁻⁴), suggesting that the model predictions for weaker interactions are also accurate. Most importantly, INDIGO performs significantly better than existing approaches, which make only qualitative predictions.

A similar problem is apparent in Fig. 6D where the observations for S. aureus are compared to the predictions: the weak correlation that is observed seems to depend entirely on 3 or 4 data points (located in the top right and bottom left). Is this correlation significant? Independent of statistical significance, it looks as if only a small fraction of the differences are correctly predicted.

The correlation is highly significant as assessed by several methods (p-value = 10⁻⁴ for spearman's correlation). Given that a sizeable fraction of the interactions are conserved between the two species, a major challenge is to identify the subset of interactions that strongly differ between them. Our results suggest that INDIGO can identify conserved interactions and those that differ between the two species. Predictions from INDIGO are both visually and quantitatively better than the correlation with experimental *E. coli* interactions.

We have expanded the validation data in this revision and our results are more significant than before (p = 10⁻⁴, r = 0.52), while the correlation with *E. coli* is only R = 0.39, further highlighting the accuracy of our approach. This result is noteworthy given that we did not use any *S. aureus* interaction data for training the model, and we also used chemogenomics data from a different species.

The usefulness of this approach is thus doubtful and needs to be clarified.

By the reviewer 1's own words "*Computational approaches for predicting drug interactions could lead to important advances in drug combination design since, if successful, they could enable the systematic exploration of the extremely large space of possible drug combinations without a need for cumbersome large scale experimental screens.*" We also would like to quote reviewer 1 to clarify the usefulness of our approach: "*The implications of a having a workable and transferable pipeline for predicting the effects of combinatorial treatments are immense -especially since this can go beyond the scope of antibiotics/adjuvants and used for drugs against other infectious and non-infectious diseases*" and "*The ability to even partially predict interactions in S. aureus with E. coli data is remarkable*".

Therefore, it is of great importance to predict interactions in clinically relevant organisms based on data from model organisms such as *E. coli*. Currently it is unclear if drug interaction outcomes in *E. coli* are conserved in other species. Here we have shown that by using chemogenomics-based models, that we can identify subsets of drug interactions that are conserved between species (broad spectrum synergy or antagonism) and those that are variable (narrow spectrum synergy or antagonism). We have identified several cases where interactions strongly differ between them (i.e., antagonistic in one species and synergistic in the other). The remarkable aspect of our approach is that only the genome information of another strain or organism is needed to extend the predictions from *E. coli* to another system. We have revised the section accordingly to further clarify the significance of our predictions.

2. The "INDIGO" algorithm is central to the results of the study. However, its description in the main text is inadequate, making it hard to assess the conceptual advance achieved in this work. This algorithm is briefly introduced in the first paragraph of the results part and in Figure 1 but the main idea and procedure used to make the predictions did not become clear to me from this. This part will be particularly hard to understand for a broad audience of biologists. The main idea behind this algorithm should be clarified and its description considerably expanded.

We agree with the reviewer and have rewritten the relevant section accordingly. We have considerably rewritten the text describing INDIGO to be understandable for a broad audience. We have also updated Figure 2 to clarify the steps involved in this process.

3. The overall approach is conceptually similar to (Yilancioglu et al., *J Chem Inf Model*, 2014) from the same last author. While this previous work focused on yeast and physicochemical properties of the drugs to predict drug interactions, the general approach of using available data on drugs and using machine learning or related approaches to predict drug interactions based on a training data set is conceptually similar. The advance of the present study thus appears incremental. It would need to be clarified how the present study presents a major step forward compared to this previous work.

The present study is fundamentally different from Yilancioglu et al. in two ways. Firstly, here we have used chemogenomics data to predict drug interaction outcomes. In contrast, Yilancioglu et al. used only chemical properties of drugs. Prediction of drug interactions using physicochemical properties is unable to offer hypotheses for mechanism of drug interactions. In our study, however, since our learning algorithm uses genetic information, genes and gene sets associated with drug interactions can be found and used for furthering the biological understanding of drug interactions.

Secondly, the Yilancioglu et al. study cannot be extended to other organisms since there is no information on species-specific effects of drug properties. However, thanks to comparative genomics and the available microbial genome sequences, our approach can be used to predict drug interactions in any species with a known genome.

4. There is little novel experimental data in this work, so the predictions made are largely about the past. E.g. the experimental results shown in Fig. 2 were generated in this study but a similar study was published almost 10 years ago (Yeh et al., *Nat Gen*, 2006) and repeated for other organisms since then (as properly acknowledged by the authors).

In our study, we report 153 and 45 drug interaction experiments in *E. coli* and *S. aureus*, respectively. (In our original submission, there were 28 experiments for *S. aureus*.) All of these data were produced in duplicate in this study. As such, our study not only has a considerable body of novel experimental data, it is on par with the drug interaction data sets reported in the literature (Yeh et al. involved 210 experiments whereas Cokol et al. involved 200). In addition, we would like to highlight that the experimental data for *S. aureus* is by far the largest existing systematic drug interaction screen in this organism, and these are all novel interactions experimentally measured for this study. Existing studies so far have focused on high-throughput screens that enhance the activity of single agents. In addition to the novel interaction dataset for *S. aureus*, a large fraction of *E. coli* data is also novel. As we had highlighted earlier, a majority of combinations (over 60%) do not overlap with Yeh et al. In addition, while Yeh et al. was a single-dose study, we have analyzed nine times more dose specific combinations than Yeh et al. while computing our drug interaction score. We have revised the results section and the abstract of the paper to highlight that these were novel experimental data sets and not extracted from literature.

Similarly, the computational analysis relies almost entirely on the published chemical genomics data set from (Nichols et al., *Cell*, 2011).

Chemogenomic profiling has been widely done for a range of model systems and an extensive resource of such data is now publicly available for *E. coli*, *S. cerevisiae*, and human cancer cell lines (Flores et al, 2005; Girgis et al, 2009; Ho et al, 2009; Lee et al, 2014; Muellner et al, 2011; Nichols et al, 2011; Tamae et al, 2008). Yet these data sets have been under-utilized for drug combination discovery. We show that one can leverage this vast resource of existing chemogenomic data to predict drug combinations for a wide range of drugs across different organisms.

It is of course perfectly fine to reanalyze existing data but the lack of new data will likely limit the appeal of this study to a more specialized community of bioinformaticians interested in drug interactions. In the absence of any mechanistic insight, the claim that drug interactions can be predicted based on chemical genomics data would be more convincing if the authors could validate predictions with a truly novel data set.

Here we have demonstrated for the first time that drug interactions in multiple pathogenic species can be predicted from chemogenomics data from model organisms. We have validated this novel approach using two novel experimental data sets: 66 new drug pairs in *E. coli* and 45 new drug pairs in *S. aureus*. Further, we demonstrate that genes that are important for drug interactions are conserved across species, hence suggesting that these mechanisms are conserved in evolution. Given the importance of multi-drug combinations across several disciplines, the novel data sets, and the provided mechanistic explanations, we believe that our paper will be relevant not only to bioinformaticians, but also to scientists interested in systems biology, drug discovery, drug resistance and infectious diseases.

One possibility could be to investigate combinations of three or more drugs. Recent studies (e.g. Wood et al., PNAS, 2012) made progress in predicting such higher order drug combinations and it would be interesting if the authors could at least discuss if their algorithm can make predictions for the effects of higher order drug combinations. It would strengthen this work if the authors could experimentally verify some of these predictions for a few selected combinations of three or more drugs.

We thank the reviewer for this suggestion. We agree that predicting higher-order combinations would be a great future application of INDIGO and we have highlighted this point in the discussion in the revised paper. Our analysis can be coupled with approaches that predict higher-order combinations based on pairwise interaction data. However, predicting pairwise combinations is still a considerable challenge, and we believe our study has taken an important step to address this issue.

2nd Editorial Decision

11 March 2016

We have now heard back from the referees who accepted to evaluate the study. Reviewer #1 saw the study for the first time, while reviewer #2 was involved in the evaluation of your first submission (reviewer #1 in previous round). As you will see, the referees raise a series of concerns, which we would ask you to convincingly address in a revision of the present work. The recommendations provided by the reviewers are very clear in this regard.

 REFEREE COMMENTS

Reviewer #1:

In this study the authors describe a computational approach INDIGO that uses *E. coli* genome wide mutant fitness data in the presence of drugs to predict drug-drug interactions. Using a training set of growth data from 15 compounds and an existing *E. coli* fitness data from >100 compounds, they identify a set of genes that can be used as classifiers to predict drug-drug interactions. Because many of these drug related genes have orthologs in diverse species, the authors used their model to predict drug interactions in two Gram-positive pathogens.

Main points:

1. Given the vast differences in gene content and physiology of *E. coli* compared to Gram-positive bacteria, I'm surprised that the *E. coli* based model made useful predictions in the other species. However, these Gram-positive predictions are only mildly accurate (Figure 5 and Supp info). The authors should discuss the limitations of their approach for predicting gene-gene interactions in non-*E. coli* species, for example INDIGO cannot model the contribution of drug related genes that are

absent in *E. coli*. The best option is to directly collect chemogenomics data in diverse species (and advances in technology will soon move us there).

2. The authors should further discuss what separates their INDIGO algorithm from existing approaches that use chemogenomics data to predict drug interactions (using global Pearson correlation). This is discussed briefly in the text but in my view (coming from a largely non-computational background), the biggest difference is that the authors take into account the individual genes with sensitive phenotypes and assign them different weights in the model. A more detailed description of what I'm missing (and readers with a similar background) would be helpful.

3. The Nichols dataset used colony imaging to assay the growth of individual mutants in the presence of drug. While this technology is not ideal for identifying detrimental genes (ie, genes that when mutated lead to an increase in fitness), there are some of these in their dataset. In addition, higher resolution approaches such as TnSeq and its associated derivatives are identifying many detrimental genes in diverse conditions. With these considerations in mind, I'm curious why the authors only looked at the sensitive genes.

4. In the first results section, the authors should give specific examples of the consistency of their results with the Yeh et al. 2006 results, ie "and was consistent with existing interaction data (Supplementary Methods)". A brief discussion of the consistency or lack thereof for certain compounds is better merited in the results, not the supplemental material. I find it a bit disturbing that the literature will now report a non-linear drug interaction for 38 compound comparisons that two groups working independently with the same compounds and same organism didn't replicate. If these drug interactions are so sensitive to growth media and methods used to monitor/measure growth, are they really going to be clinically relevant?

5. If the authors want to really claim that INDIGO makes quantitative predictions of synergy and antagonism, then they should plot the actual measured and predicted interaction scores in Figure 3, not the ranks.

6. The focus on a small set of "DIR" drug-interaction related genes seems the wrong approach to identifying novel antimicrobial targets and modes of action. It also biases the search for synergistic interactions to a limited set of cellular processes and likely ignores other important ways in which drug-gene interactions could occur. The authors should at least mention this in the discussion.

7. Related to the previous point, more attention should be given to explaining the mechanistic underpinnings of specific drug-interactions outcomes and why the predictions likely worked or failed in the Gram-positive or Mycobacterial systems. At the end of the day an $R = 0.52$ is not great, and it would be nice to know which compounds are consistently working or failing.

8. The tendency of the authors to use summary figures without drug mechanism and even drug names in many cases obfuscates the method. I think the paper would be much stronger with more mechanistic discussion and less focus on scoring the algorithms predictive power. Ultimately, computational methods such as this are valuable for moving beyond a set of compounds for which a wealth of experimental data exists and drug-interaction data can be readily and expediently obtained. What about a 100x100 matrix of compounds including new screening hits?

Minor points:

1. In Figure 1, add a color bar legend from blue to red that indicates the drug interaction scores.
2. In Figure 1 legend, specify that the rank correlation is used, rather than the generic "correlation".
3. The reproducibility of the drug interaction scores (and presumably the growth data they are based on) is not that great. The authors should state why this is the case and discuss whether more sensitive growth assays or more biological replicates would help. This is especially important in the light of these types of data being used for ultimately making clinical decisions.
4. I suggest making Figure 1 more compelling and easier to follow by labeling the compounds (at least on the y-axis) based on mode of action or structure.
5. Remove or replace "Remarkably" from the second paragraph of the results section "Experimental validation of novel predictions for 66 drug-drug interactions"

6. The authors should give an example or two of drugs with similar mechanisms of actions and/or chemogenomic profiles having synergistic or antagonistic outcomes (in the Experimental validation of novel predictions for 66 drug-drug interactions" results section.
7. How many Drug Interaction Related (DIR) genes are there total?
8. I don't understand what is being plotted in Figures 5D and 5E. Why are these rank ordered and not just the actual predicted interaction values?
9. In the Figure 5DE legends, it described experimentally measured interactions but the plot shows predicted interaction values. Also, as far as I can tell, no experimentation was done on *M. tuberculosis*.
10. It's unclear how the growth assays were done. In a microplate reader?
11. I don't understand why the same "area under the curve" methods were not used for both *E. coli* and *S. aureus*

Reviewer #2:

The revised manuscript by Chandrasekaran et al. fixes a number of flaws of the first submission and addresses the majority of the comments raised by both reviewers. Overall I find the revised manuscript to be better structured, methods and figs are better explained, messages are better conveyed and focus is kept on its strong points. As mentioned in first round, I find both the approach novel and the topic highly interesting. Nevertheless, and given the time passed from earlier submission (see also point 1), in my opinion the authors have to address a few more points on their manuscript - all can be fixed by small edits/additions.

Major points

1. In a recently published paper by the labs of Mike Tyers and Gerry Wright (Cell Systems, 2016) chemical genetics data and structural features of drugs are used to train a predictor (machine learning) for species-specific interactions. Although methods used and angle is rather different from this manuscript, the authors should discuss this paper: commonalities, complementarity of approaches but also some of the differences in the findings - e.g. Wildehain et al. find that chemical structural features of drugs have to be included for predictor to work.
2. The authors should acknowledge in the manuscript that they see about twice as many interactions (56 antagonisms + 14 synergies) as non-interactions (35 neutral interactions), and discuss why this happens. I can understand the claims for more antagonism than synergy, but interactions being more common than non-interactions defy the null hypothesis of any interaction model, and should be addressed.
3. In the resubmission authors increased the # of drug-drug interactions tested for *S. aureus* (2 new drugs are tested and a 10 x 10 matrix is presented). Correlation between experimentally tested interactions in *S. aureus* & *E. coli* drops from 0.48 in previous 8 x 8 dataset to 0.39 (Fig S17A). When incorporating the Indigo Predictions correlation goes up to 0.47 (Fig S5D and S17B; this analysis is missing from first submission and 8 x 8 dataset), which I would say is a marginal improvement - and comes from including more drugs that are predicted to behave differently between 2 organisms and thus need the Indigo predictions. So although I don't disagree with authors' statement that Indigo helps with predicting the interactions between organisms, I think the authors should make it clear that this about as good as it gets at the moment and still most of the "predictive power" comes from conserved interactions. So I would tone down abstract ("successfully predicted to some degree") and leave open the room for improving predictions across species in the future in discussion.

Minor points

1. Intro: define/cite the study you use for the chemogenomics data (Nichols et al.)
2. In Fig 1 authors show the average AUC between the 2 replicates for each of the drug-pair concentration. Yet they don't use these data for calculating the final interaction α score, but instead, they calculate α scores on each experiment separately and average them. Although I don't disagree with the strategy, especially given the # of replicates, authors should make this clear in text. This is the reason for the discrepancy between some of the checkerboard patterns and the way interaction is called in Fig 1.

3. Interaction quantification; how did authors calculate largest isophenotypic contour for combinations that only 2 or 3 drug concentrations gave measurable growth?
4. Chemogenomic profile: score compares fitness of mutant in a treatment to both wildtype fitness under same treatment and mutant fitness under no treatment. This is why it is actually not a z score.
5. INDIGO: although authors refer to Methods for explaining the random forest algorithm, there is no further mention to this in text. Reply to reviewer 1 as to inputs on INDIGO can be included in text as it provides some more relevant info.
6. Figs 3E and 3F are not mentioned in txt
7. INDIGO predicts better synergies than antagonisms (ROC curves), despite being built with more antagonisms. This is not pointed out/discussed, despite authors mentioning in rebuttal that it is in Results & Discussion. Either I missed it or authors did.
8. Benchmarking with Yeh data: supplementary scatter plots should be provided in addition to description in text.
9. Peroxide response and discrepancy from Brynildsen (2013). According to Table 1, H₂O₂ has an MIC of 250 µg/ml (~7.34 mM - which at the high side), and you use sub-MIC concentrations. 5 mM is used in the Brynildsen study, so concentration cannot be the reason. Not sure why duration of treatment would make a difference either.
10. Table 2b; needs correction for multiple testing
11. Not clear how enrichment in DIR genes is concordant/related to Chevereau & Bollenbach study. Authors should elaborate more on this.
12. Staph data (Fig S13) - include plots for replicate correlation (mentioned in response to review 1).
13. Mtb interaction scores used from literature are not quantitative - should be pointed out.
14. There is still no reference in main text for some of the Supp Material.

1st Revision - authors' response

09 April 2016

Reviewer #1:

In this study the authors describe a computational approach INDIGO that uses E. coli genome wide mutant fitness data in the presence of drugs to predict drug-drug interactions. Using a training set of growth data from 15 compounds and an existing E. coli fitness data from >100 compounds, they identify a set of genes that can be used as classifiers to predict drug-drug interactions. Because many of these drug related genes have orthologs in diverse species, the authors used their model to predict drug interactions in two Gram-positive pathogens.

Main points:

1. Given the vast differences in gene content and physiology of E. coli compared to Gram-positive bacteria, I'm surprised that the E. coli based model made useful predictions in the other species. However, these Gram-positive predictions are only mildly accurate (Figure 5 and Supp info). The authors should discuss the limitations of their approach for predicting gene-gene interactions in non-E. coli species, for example INDIGO cannot model the contribution of drug related genes that are absent in E. coli. The best option is to directly collect chemogenomics data in diverse species (and advances in technology will soon move us there).

We agree with the reviewer that there is still room for improvement in terms of predictions, and the predictions for other organisms would be more accurate with chemogenomics data for the corresponding strains. New technologies for the generation of chemogenomic data sets for pathogens will undoubtedly move us towards more reliable prediction of drug interactions in pathogens. So far however, we have not yet generated such data for relatively well-studied pathogens such as *S. aureus* or *M. tuberculosis*. INDIGO thus provides a first estimate of drug interactions in these circumstances. In general, our discovery that we can predict drug interactions in other species by mapping orthologous genes thus provides a mechanism for extending predictions from model organisms or non-pathogenic strains to less-studied pathogens.

The predictive power of INDIGO is also dependent on the drug interaction experiments conducted. Therefore, performing drug interaction assays in pathogens for training INDIGO is another venue that will improve INDIGO predictions.

We have revised the Discussion section of the paper to highlight this important point raised by the reviewer as follows (Page 11, last paragraph) –

“While this capability to predict drug interactions by mapping orthologous genes is a significant advance for this field and can be extended to any organism with genome sequence information, a limitation of this approach is that INDIGO cannot explicitly model the contribution of pathogen DIR genes that are not present in *E. coli*. Our interaction estimates for these systems could be further improved with the availability of chemogenomics data for these pathogens and by performing drug interaction measurements directly in pathogens as training data for INDIGO. Predictive accuracy of INDIGO can be enhanced in the future through new technologies for accurate measurement of drug-drug and chemical-genetic interactions, performing chemogenomic screens with essential genes (Cameron & Collins, 2014), and by harnessing drug physicochemical properties and chemical structure (Wildenhain et al, 2015). INDIGO can complement theoretical models for predicting multi-drug interactions from pairwise interactions (Wood et al, 2012), and kinetic modeling-based approaches, which are currently restricted to small pathways due to a lack of known kinetic parameters and drug targets (Singh et al, 2015). By transforming drug responses and interactions into the genomic space, our study provides a framework for genomics-driven drug combinations discovery. “

2. The authors should further discuss what separates their INDIGO algorithm from existing approaches that use chemogenomics data to predict drug interactions (using global Pearson correlation). This is discussed briefly in the text but in my view (coming from a largely non-computational background), the biggest difference is that the authors take into account the individual genes with sensitive phenotypes and assign them different weights in the model. A more detailed description of what I'm missing (and readers with a similar background) would be helpful.

As the reviewer correctly points out, the biggest difference is that INDIGO take into account the individual genes with sensitive phenotypes and quantifies their contribution to the drug-interaction outcome. In contrast, existing approaches use a bulk aggregate score to quantify the chemogenomic profile similarity. Hence, instead of using just one predictor (aggregate correlation or similarity), we are using hundreds of predictors in our study.

Another important difference is that, the contribution of each gene is contextual, i.e. depends on the state of other genes (Figure 2). We have revised the Introduction, Results and Introduction sections to highlight the main difference of our approach compared to other existing approaches as follows –

[Page 6, 2nd paragraph]

“INDIGO thus takes a systems approach to predict drug interactions; INDIGO quantifies the contribution of individual genes with a chemical-genetic interaction on the overall drug interaction outcome (synergy or antagonism). Importantly, the contribution of each gene in INDIGO is contextual, i.e. depends on the state of other genes.”

[Page 4, 2nd paragraph]

“INDIGO greatly expands the capability of current drug interaction prediction approaches by estimating interaction outcomes in clinically relevant pathogens by finding orthologous genes in model organisms.”

[Page 11, 1st paragraph]

“INDIGO assesses the influence of hundreds of individual chemical-genetic interactions on drug-drug interaction outcomes. In contrast, existing chemogenomics-based approaches determine synergy based on a single aggregate metric of drug similarity. Our gene-centric model has also enabled us to apply our *E. coli* drug interaction model to predict outcomes in other bacterial species.”

3. The Nichols dataset used colony imaging to assay the growth of individual mutants in the presence of drug. While this technology is not ideal for identifying detrimental genes (ie, genes that when mutated lead to an increase in fitness), there are some of these in their dataset. In addition, higher resolution approaches such as TnSeq and its associated derivatives are identifying many detrimental genes in diverse conditions. With these considerations in mind, I'm curious why the authors only looked at the sensitive genes.

According to Nichols et al, “Overall, 80% of the phenotypes were negative (gene deletion more sensitive) and 20% positive (gene deletion more resistant), consistent with recent genetic interaction analyses in *S. cerevisiae* (Fiedler et al., 2009) and *S. pombe* (Roguev et al., 2008). This suggests that removal of a gene product is more likely to decrease than enhance resistance to stress (Figure S2B).” Hence, despite these biases pointed out by the reviewer, sensitive interactions are statistically more abundant and provide more information to the model than resistant interactions. As more effective screens become available in the future that avoid these biases, we expect that resistant gene data can also be used in building the model. We have highlighted this issue in the appendix (section: chemogenomics data processing; see below)

[Appendix, Page 3, 3rd paragraph]

In this study, we only used genes that lead to increased sensitivity (and not resistance) in the chemogenomics data. As a result of the Nichols et al study design, there were lot more statistically significant associations for sensitivity than resistance. 80% of the reported phenotypes were negative (gene deletion more sensitive) and 20% positive (gene deletion more resistant), consistent with other recent chemical-genetic interaction analyses in *S. cerevisiae* and *S. pombe*. Hence, only gene sensitivity profiles were used as they were more abundant and statistically significant in Nichols et al. In the future, it should be possible to include resistant genes in the INDIGO framework.

4. In the first results section, the authors should give specific examples of the consistency of their results with the Yeh et al. 2006 results, ie "and was consistent with existing interaction data (Supplementary Methods)". A brief discussion of the consistency or lack thereof for certain compounds is better merited in the results, not the supplemental material. I find it a bit disturbing that the literature will now report a non-linear drug interaction for 38 compound comparisons that

two groups working independently with the same compounds and same organism didn't replicate. If these drug interactions are so sensitive to growth media and methods used to monitor/measure growth, are they really going to be clinically relevant?

We completely agree with the reviewer that this is an important issue – this was the rationale for doing a more comprehensive experimental screen that looks across multiple doses in this study. We believe that our improved experimental setup that looks at a larger dose range (rather than single dose in Yeh et al.) should be better representative of the drug interaction outcomes.

Nevertheless, only one combination among the 66 interactions disagrees completely (i.e. synergistic in one data set and antagonistic in the other). We have also included a supplementary figure summarizing the similarities and differences between the two data sets (Appendix Figure S3) in the revised paper.

Therefore, while it is heartening to see broad similarities in the distribution of the interaction outcomes in Yeh et al and our study, it is not a direct validation of our approach due to differences in media and dosage used in the studies.

5. If the authors want to really claim that INDIGO makes quantitative predictions of synergy and antagonism, then they should plot the actual measured and predicted interaction scores in Figure 3, not the ranks.

We have plotted the predictions for all the three organisms with actual values as an expanded view figure (Figure EV3; shown below). The reason for plotting ranks was to show that the model was robust to outliers in the data and the method of normalization. Our correlations are in fact higher with actual values ($R = 0.57$, $p\text{-value} = 10^{-7}$) compared to rank-transformed values ($R = 0.52$, $p\text{-value} = 10^{-6}$). Rank transformed data are also quantitative and more accurate than the actual values as they are robust to outliers in the data. In the revised paper, we have emphasized the rationale for using rank-normalized data in the figure legends.

Expanded view figure 3: Scatter plots of measured interaction scores and predictions by INDIGO with drug combination labels (raw scores). We have used rank normalized data for visualization in Figure 3 and 5, and for quantifying the accuracy of predictions as they are robust to outliers and the method of normalization. The correlations are higher with actual values (Pearson correlation $R = 0.57$, $p\text{-value} = 10^{-7}$) compared to rank transformed values ($R = 0.52$, $p\text{-value} = 10^{-6}$) for *E. coli* predictions. Strong synergistic, antagonistic interactions and outliers are highlighted for *E. coli* (panel A), *S. aureus* (panel B; rank correlation $R = 0.47$; Pearson correlation $R = 0.5$) and *M. tuberculosis* (panel C; rank & Pearson correlation $R = 0.54$) (*M. tuberculosis* interaction data are qualitative in nature). Abbreviations: NIG – Nigericin, STR – Streptomycin, CLA – Clarithromycin, MIT – Mitomycin. Please refer to table 1 for full list of drugs and abbreviations.

6. The focus on a small set of "DIR" drug-interaction related genes seems the wrong approach to identifying novel antimicrobial targets and modes of action. It also biases the search for synergistic interactions to a limited set of cellular processes and likely ignores other important ways in which drug-gene interactions could occur. The authors should at least mention this in the discussion.

By focusing on the top DIR, our aim was to identify the dominant pathways that were commonly used by many drugs. However, this was not meant to be exhaustive, and as predicted by our approach, drug interactions are predicted to involve contributions from hundreds of genes involving a range of cellular processes. We agree that by focusing only on the top DIR genes we might miss other important mechanisms of interaction. We have addressed this rationale and concern in the results section as follows –

[Page 8, 3rd paragraph]

“Overall, the interaction outcomes for each drug combination depends on complex interaction between many genes. Nevertheless, the presence of these top genes can be a strong predictor of synergy or antagonism. For example, we find genes associated with synergy to be enriched in the chemogenomic profile of triclosan (p-value = 10^{-10} , hypergeometric test), explaining its promiscuous synergy. By ranking genes used by INDIGO, we can determine the dominant pathways that are commonly used across drug classes.”

We have also included the entire ranked list of DIR genes as a supplementary table (supplementary dataset 3) to enable future discovery of drug-interaction mechanisms.

7. Related to the previous point, more attention should be given to explaining the mechanistic underpinnings of specific drug-interactions outcomes and why the predictions likely worked or failed in the Gram-positive or Mycobacterial systems. At the end of the day an $R = 0.52$ is not great, and it would be nice to know which compounds are consistently working or failing.

We thank the reviewer for the comment. We have addressed this issue in two ways as follows –

1. For the E. coli predictions we have identified compounds for which the algorithm consistently works well based on leave-one-drug-out cross validation (S. Figure 8). Through this analysis we found that INDIGO can accurately predict interactions for most compounds and identified hydrogen peroxide as the compound for which INDIGO makes the most errors in prediction. The inability to predict hydrogen peroxide interactions suggest that its mechanism of action is different from what is learned from its chemogenomic profile. We have updated the Results section of the paper to highlight the drugs that lead to incorrect predictions as follows –

[Page 7, 3rd paragraph]

“Cross-validation analysis also revealed that predictions were consistently accurate across most drugs and identified hydrogen peroxide as leading to the largest errors in predictions (Appendix Figure S8, S9). This suggests that the chemogenomics profile of hydrogen peroxide may not accurately reflect its mechanism of interaction.”

2. For cross-species predictions, we have quantified for both M.tb and S. aureus compounds that are more likely to have conserved interactions. For specific cases such as tetracycline, we have explained the reason the model predictions succeeded or failed in predicting interaction outcomes based on conservation of the drug-interaction-related genes. We have

now included this analysis of drug-interaction conservation as an expanded view figure (Figure EV4; shown below) instead of in the supplement.

Expanded view figure 4: Interaction conservation between *E. coli* and *S. aureus* (top panel) and *E. coli* and *M. tuberculosis* (bottom panel). The heatmaps summarize the interaction outcome between drugs from similar or different chemical families (left panel) and target processes (right panel). The heat-maps represents the entire drug interaction matrix, inferred from the compendium of Nichols et al., collapsed into 13 major drug families and 10 major target processes. To determine average conservation of interactions between different groups, we compared the differences in interaction scores between the two species for all drug combinations between two groups (drug family or target process) with the background interaction score difference for all drug pairs.

8. *The tendency of the authors to use summary figures without drug mechanism and even drug names in many cases obfuscates the method. I think the paper would be much stronger with more mechanistic discussion and less focus on scoring the algorithms predictive power. Ultimately, computational methods such as this are valuable for moving beyond a set of compounds for which a wealth of experimental data exists and drug-interaction data can be readily and expediently obtained. What about a 100x100 matrix of compounds including new screening hits?*

We thank the reviewer for these suggestions. We have updated several sections of the manuscript with mechanistic information related to the drugs mechanism of action and interaction (described below):

1. We have updated the figures (1, 3 & 5) to include more mechanistic information regarding target processes of the antibiotics (updated Figure 1 shown below).
2. We have also provided information on interaction outcomes for antibiotics targeting similar and diverse target processes (results section; see minor comment #6).

3. We have added expanded view figures that have the drug names on the scatter plots (Figure EV3; refer comment #5).
4. We have included more mechanistic discussions on specific cases related to the interaction outcome based on the presence of certain genes in the chemogenomic profile of these drugs. Overall, the interaction outcomes for each drug combination are quite complex and depend on many genes. Nevertheless, the presence of certain genes can predispose to synergy or antagonism. For example, we find genes associated with synergy to be enriched in the profile of triclosan. This explains the promiscuous synergy of triclosan. We have updated the Results section of the paper to reflect this as follows –

[Page 8, 3rd paragraph]

“The presence of these DIR genes in the sensitivity profile of the drugs also correlated with either synergy or antagonism (Table 2a). Overall, the interaction outcomes for each drug combination depends on complex interaction between many genes. Nevertheless, the presence of these top genes can be a strong predictor of synergy or antagonism. For example, we find genes associated with synergy to be enriched in the chemogenomic profile of triclosan, explaining its promiscuous synergy. By ranking genes used by INDIGO, we can determine the dominant pathways that are commonly used across drug classes.”

5. By looking at the conservation of this gene set, we can also explain the variability of tetracycline and quinolones (see comment #7).

We agree that our approach is currently restricted to compounds with chemogenomics data. Yet, chemogenomic screens can now be done quickly and efficiently for a large number of model systems. In our manuscript we have performed predictions for 73 drugs and 53 stress agents; the 126 x 126 matrix of interactions is provided as supplementary dataset 2.

Similarly, screening hits from a 100x100 set of uncharacterized compounds could subsequently be profiled using chemogenomics. The interactions of the screening hit compounds with hundreds of existing drugs can then be quantified using INDIGO.

Figure 1 (shown above), Figure 3 and Figure 5 have been updated to have cartoons depicting the mechanism of action of the antibiotics.

Minor points:

1. In Figure 1, add a color bar legend from blue to red that indicates the drug interaction scores.

2. In Figure 1 legend, specify that the rank correlation is used, rather than the generic "correlation".

We have updated Figure 1 accordingly (see above).

3. The reproducibility of the drug interaction scores (and presumably the growth data they are based on) is not that great. The authors should state why this is the case and discuss whether more sensitive growth assays or more biological replicates would help. This is especially important in the light of these types of data being used for ultimately making clinical decisions.

The correlation between replicates is very high ($R = 0.81$, $p\text{-value} = 10^{-26}$). While there is some variability between replicates in terms of degree of synergy or antagonism, in terms of interaction outcomes none of 171 combinations showed a change from synergy to antagonism between replicates. Overall we do agree with the reviewer that more sensitive growth assays and more biological replicates would help increase the correlation between replicates further. We have updated the Discussion section accordingly as follows –

[Page 11, 4th paragraph]

“Predictive accuracy of INDIGO can be enhanced in the future through new technologies for accurate measurement of drug-drug and chemical-genetic interactions, performing chemogenomic screens with essential genes (Cameron & Collins, 2014), and by harnessing drug physicochemical properties and chemical structure (Wildenhain et al, 2015).”

4. I suggest making Figure 1 more compelling and easier to follow by labeling the compounds (at least on the y-axis) based on mode of action or structure.

We thank the reviewer for the suggestion. We have now added a small image next to the drugs that show the cellular process targeted by each drug (see Figure 1 above comment #8).

5. Remove or replace "Remarkably" from the second paragraph of the results section "Experimental validation of novel predictions for 66 drug-drug interactions"

We have updated the text and removed “Remarkably”.

6. The authors should give an example or two of drugs with similar mechanisms of actions and/or chemogenomic profiles having synergistic or antagonistic outcomes (in the Experimental validation of novel predictions for 66 drug-drug interactions" results section.

We thank the reviewer for the suggestion. We found that drugs with similar targets and chemogenomic profiles can have both synergistic and antagonistic outcomes. For example, combinations of tobramycin-spectinomycin, tobramycin - gentamicin, and fusidic acid - clarithromycin, share similar chemogenomic profiles and target processes, yet have antagonistic, neutral and synergistic outcomes respectively (Figure EV1; see below). We have updated this section accordingly and added a new expanded view figure showing the chemogenomic similarity scores of these combinations.

We have updated the Results section of the paper as follows:

[Page 7, 2nd paragraph]

“Our data shows that drugs with similar targets and chemogenomic profiles can have both synergistic and antagonistic outcomes. For example, combinations of tobramycin-spectinomycin, tobramycin - gentamicin, and fusidic acid - clarithromycin, share similar chemogenomic profiles and target processes, yet have antagonistic, neutral and synergistic outcomes respectively (Figure EV1).”

Expanded view figure 1: Drugs with similar targets and chemogenomic profiles can have both synergistic and antagonistic interactions. Combinations highlighted in this plot tobramycin - spectinomycin, tobramycin - gentamicin, and fusidic acid - clarithromycin, share similar chemogenomic profiles, as measured by the correlation between their chemogenomic profiles, and similar target processes. Yet they have antagonistic, neutral or synergistic outcomes suggesting that chemogenomic profile or target similarity is not a strong predictor of drug interaction outcome.

7. How many Drug Interaction Related (DIR) genes are there total?

The Supplementary dataset 3 has the full list of genes and the corresponding rank by INDIGO. We defined the set of top 250 genes that contributed to over 75% of the variance predicted by the model as Drug Interaction Related (DIR) genes. We have updated the ‘**Genetic predictors of synergy and antagonism**’ section as follows –

[Page 8, 2nd paragraph]

“The top 81 genes accounted for 50%, the top 222 accounted for 75%, and the top 581 accounted for 95% of variance in the predicted data (Figure EV3). We defined the set of top 250 genes that contributed to over 75% of the variance predicted by the model as Drug Interaction Related (DIR) genes.”

8. I don't understand what is being plotted in Figures 5D and 5E. Why are these rank ordered and not just the actual predicted interaction values?

We had used ranks for reasons mentioned earlier with respect to robustness to outliers (refer comment #5); we have also now plotted the same using actual values as an enhanced view figure (Figure EV3). The correlation is once again higher with actual values; we nevertheless recommend using rank normalized data.

9. In the Figure5DE legends, it described experimentally measured interactions but the plot shows predicted interaction values. Also, as far as I can tell, no experimentation was done on *M. tuberculosis*.

We have updated the figure legend and results section to clarify that these data were obtained from the literature. Further, we have updated the legend to clarify that the x-axis shows model predictions and the y-axis shows literature data for *M. tb*.

We have updated Figure 5E legend as follows:

“*M. tuberculosis* interaction data were curated from literature and are qualitative in nature. See Figure EV3 for a more detailed plot with drug labels.”

10. *It's unclear how the growth assays were done. In a microplate reader?*

We used TECAN Infinite F200 microplate reader for our growth assays. We have updated the Methods section accordingly as follows

[Page 12, 3rd paragraph]

“For *E. coli* experiments, optical density (OD) measurements were done every 15 minutes for 12 hours in a TECAN Infinite F200 microplate reader. Growth rate was then estimated based on the area under the growth curve.”

11. *I don't understand why the same "area under the curve" methods were not used for both E. coli and S. aureus*

From our *E. coli* experimental data, we found that the interaction measures from end point and AUC methods were highly correlated. Since end point readings are faster and easier to measure (two factors that are especially important when handling potentially pathogenic bacteria), we used the end-point measurements for *S. aureus*.

Reviewer #2:

The revised manuscript by Chandrasekaran et al. fixes a number of flaws of the first submission and addresses the majority of the comments raised by both reviewers. Overall I find the revised manuscript to be better structured, methods and figs are better explained, messages are better conveyed and focus is kept on its strong points. As mentioned in first round, I find both the approach novel and the topic highly interesting. Nevertheless, and given the time passed from earlier submission (see also point 1), in my opinion the authors have to address a few more points on their manuscript - all can be fixed by small edits/additions.

Major points

1. In a recently published paper by the labs of Mike Tyers and Gerry Wright (Cell Systems, 2016) chemical genetics data and structural features of drugs are used to train a predictor (machine learning) for species-specific interactions. Although methods used and angle is rather different from this manuscript, the authors should discuss this paper: commonalities, complementarity of approaches but also some of the differences in the findings - e.g. Wildenhain et al. find that chemical structural features of drugs have to be included for predictor to work.

As the reviewer correctly suggests, these studies are complementary in several aspects. While the Tyers-Wright manuscript focuses on finding species specific interactions experimentally, our approach uses a computational approach to discover these outcomes. Their data set and analysis are hence complementary to our study and could be used for developing drug interaction models for yeast species. We have cited this study in the Discussion section as follows –

[Page 11, 3rd paragraph]

“Species-specific interaction outcomes have also been observed for drug combinations against fungal pathogens, similar to our observation in bacteria (Wildenhain et al, 2015).”

Another complementary area of these studies is the use of compound structure to make predictions. Although they are less accurate than the chemogenomics-based predictions reported in our study, combining these approaches can potentially lead to enhanced predictive power. We have cited this study in the Discussion section as follows –

[Page 11, 4th paragraph]

“Predictive accuracy of INDIGO can be enhanced in the future through new technologies for accurate measurement of drug-drug and chemical-genetic interactions, performing chemogenomic screens with essential genes (Cameron & Collins, 2014), and by harnessing drug physicochemical properties (Yilancioglu et al, 2014) and chemical structure (Wildenhain et al, 2015).”

2. The authors should acknowledge in the manuscript that they see about twice as many interactions (56 antagonisms + 14 synergies) as non-interactions (35 neutral interactions), and discuss why this happens. I can understand the claims for more antagonism than synergy, but interactions being more common than non-interactions defy the null hypothesis of any interaction model, and should be addressed.

Our null hypothesis based on the Loewe’s interaction model is that a drug is non-interacting with itself. Deviations from this null model leads to either synergy or antagonism. Hence using this framework, it is possible that there are considerably more antagonistic interactions than non-interactions. There is no underlying assumption that the data should be normally distributed. We have updated the Methods section to make our null hypothesis clear as follows –

[Page 12, 2nd paragraph]

“Our null hypothesis based on the Loewe’s interaction model is that a drug is non-interacting with itself. Deviations from this null model leads to either synergy or antagonism. The advantage of this approach is that there is no underlying assumption that the data should be normally distributed with similar numbers of synergy and antagonism, or that neutral interactions should be more common than synergy or antagonism.”

Interestingly, consistent with our null hypothesis, INDIGO correctly predicted no interaction between the same drug at different doses, with each dose represented by a unique chemogenomic profile (median score = 0 for self-self interactions involving 73 drugs; Appendix Figure S16). This indicates that INDIGO can differentiate chemogenomic profiles of the same drug at different doses from other drugs.

3. In the resubmission authors increased the # of drug-drug interactions tested for S. aureus (2 new drugs are tested and a 10 x 10 matrix is presented). Correlation between experimentally tested interactions in S. aureus & E. coli drops from 0.48 in previous 8 x 8 dataset to 0.39 (Fig S17A). When incorporating the Indigo Predictions correlation goes up to 0.47 (Fig S5D and S17B; this analysis is missing from first submission and 8 x 8 dataset), which I would say is a marginal improvement - and comes from including more drugs that are predicted to behave differently between 2 organisms and thus need the Indigo predictions. So although I don't disagree with authors' statement that Indigo helps with predicting the interactions between organisms, I think the authors should make it clear that this about as good as it gets at the moment and still most of the "predictive power" comes from conserved interactions. So I would tone down abstract ("successfully predicted to some degree") and leave open the room for improving predictions across species in the future in discussion.

We agree with the reviewer that there is more room for improvement and have updated the language in the manuscript. We have revised the Abstract and replaced “predicted” with “estimated”. While we cannot accurately predict the exact outcome in other species, a key strength is that we can still accurately predict the combinations that would be conserved or likely to change between the species based on the deviation score. We have also updated the Discussion to reflect this as follows –

[Page 11, 4th paragraph]

“While this capability to predict drug interactions by mapping orthologous genes is a significant advance for this field and can be extended to any organism with genome sequence information, a limitation of this approach is that INDIGO cannot explicitly model the contribution of DIR genes that are not present in *E. coli*. Our interaction estimates for these systems could be further improved with the availability of chemogenomics data for these pathogens and by performing drug interaction measurements directly in pathogens as training data for INDIGO.”

Minor points

1. Intro: define/cite the study you use for the chemogenomics data (Nichols et al.)

We have revised the text accordingly.

2. In Fig 1 authors show the average AUC between the 2 replicates for each of the drug-pair concentration. Yet they don't use these data for calculating the final interaction α score, but instead, they calculate α scores on each experiment separately and average them. Although I don't disagree with the strategy, especially given the # of replicates, authors should make this clear in text. This is the reason for the discrepancy between some of the checkerboard patterns and the way interaction is called in Fig 1.

We have revised the text accordingly. We have made it clear that the final α scores were estimated separately for each replicate and then averaged. We have revised Figure 1 legend as follows –

“The α -scores were calculated for each replicate and the average score was overlaid on the growth data.”

3. Interaction quantification; how did authors calculate largest isophenotypic contour for combinations that only 2 or 3 drug concentrations gave measurable growth?

We defined the largest isophenotypic contour as the largest unbroken contour that connects drug 1 to drug 2. This is the identical definition used previously in Cokol et al 2011 MSB. This contour has the largest number of data points and is used for fitting to the function to compute α score. This quantification method is most robust when the inhibition level in the largest isophenotypic contours is reached at the largest doses used, since then isophenotypic contours will be connecting two far edges of the checkerboard matrix. However, for the quantification to work, the minimum requirement is some inhibition at the lowest dose of a drug used. Therefore, even if MIC is prematurely reached at the third dose used, interaction is still quantifiable.

4. Chemogenomic profile: score compares fitness of mutant in a treatment to both wildtype fitness under same treatment and mutant fitness under no treatment. This is why it is actually not a z score.

We thank the reviewer for pointing this out. We have revised the text and call the score as chemogenomics fitness-score instead of z-score.

5. *INDIGO*: although authors refer to Methods for explaining the random forest algorithm, there is no further mention to this in text. Reply to reviewer 1 as to inputs on *INDIGO* can be included in text as it provides some more relevant info.

We have revised the results section as follows –

[Page 6, 1st paragraph]

“A machine learning algorithm called random-forests is then used to build a predictive model that links interaction outcome of drug combinations to the joint chemogenomic profile of the drug pair (Methods; Figure 2). The random forest algorithm builds an ensemble of decision trees using the training data set and outputs the mean prediction of the individual trees; it also identifies genes in the chemogenomics data that are most predictive of drug interactions. *INDIGO* learns the mechanism of drug interactions from the chemogenomics data in an unsupervised fashion by using the random forest algorithm.”

6. *Figs 3E and 3F are not mentioned in txt*

We apologize for the oversight. We have revised the Results and Methods section to mention these figures

7. *INDIGO predicts better synergies than antagonisms (ROC curves), despite being built with more antagonisms. This is not pointed out/discussed, despite authors mentioning in rebuttal that it is in Results & Discussion. Either I missed it or authors did.*

INDIGO performs equally well for both synergy and antagonism based on area-under-the-curve measurement of the ROC curves - AUC for synergy = 0.79, p-value = 10^{-16} ; AUC for antagonism = 0.8, p-value = 10^{-16} , as highlighted in the Methods section. Correlation or similarity based approaches perform poorly for antagonism (based on the ROC curves) because they do not have a model for predicting antagonism. We have also highlighted the actual AUC values in the legend of the ROC curves.

We had misinterpreted this comment as relating to the differences in distribution of the training data set which we had addressed in the the methods section – “controls for *INDIGO*” as it fit better in that section. The theoretical framework of the random forest algorithm allows it to be robust to these differences in the distribution of the training set.

8. *Benchmarking with Yeh data: supplementary scatter plots should be provided in addition to description in text.*

We have added a supplementary figure that compares the interaction scores in both the data sets (Appendix Figure S3).

9. *Peroxide response and discrepancy from Brynildsen (2013). According to Table 1, H₂O₂ has an MIC of 250 µg/ml (~7.34 mM - which at the high side), and you use sub-MIC concentrations. 5 mM is used in the Brynildsen study, so concentration cannot be the reason. Not sure why duration of treatment would make a difference either.*

We believe that one important reason for this discrepancy could be due to the difference in intracellular peroxide generation (the focus of Brynildsen et al study) and extracellular addition of peroxide (done in our study). Dwyer et al recently highlighted that the levels of intracellular and extracellular peroxide can be very different, and not reach equilibrium, due to biological constraints on H₂O₂ diffusion across the membrane, compartment-specific scavenging of H₂O₂, and rapid Fenton chemistry destruction of intracellular H₂O₂ (Dwyer et al, 2014). Brynildsen et al predicted

synergy with bactericidal antibiotics with gene knockouts that lead to the generation of intracellular peroxide. The extracellular addition of H₂O₂ in our study may not hence have similar effect as intracellular generation of H₂O₂. We overall agree that this is a surprising observation and needs further analysis. We have updated the Methods section discussing the surprising effect of hydrogen peroxide as follows –

[Appendix, Page 2, 4th paragraph]

‘We also observed very strong antagonism of hydrogen peroxide with other antibiotics, not observed in a previous study (Brynildsen et al, 2013). Brynildsen et al predicted synergy between bactericidal antibiotics and gene knockouts that lead to the generation of intracellular hydrogen peroxide. Dwyer et al recently highlighted that the levels of intracellular and extracellular hydrogen peroxide can be very different, and not reach equilibrium, due to biological constraints on hydrogen peroxide diffusion across the membrane, compartment-specific scavenging of hydrogen peroxide, and rapid Fenton chemistry destruction of intracellular hydrogen peroxide. The extracellular addition of hydrogen peroxide in our study may not have a similar effect as intracellular generation. At this high concentration, hydrogen peroxide has been observed to be bacteriostatic (Imlay, 2015) and hence may lead to antagonism with many antibiotics, consistent with other studies (Lobritz et al, 2015; Ocampo et al, 2014).’

10. Table 2b; needs correction for multiple testing

We have revised Table 2b with p-values corrected for multiple hypotheses testing

11. Not clear how enrichment in DIR genes is concordant/related to Chevereau & Bollenbach study. Authors should elaborate more on this.

We have revised our text to better explain the connection with Chevereau & Bollenbach study as follows –

[Page 8, 3rd paragraph]

“Notably, lipopolysaccharide biosynthesis and oxidative phosphorylation were found as two of the top ten pathways associated with drug interactions. This is in agreement with a recent study that screened antibiotic combinations in all non-essential gene deletion strains of *E. coli* and found that strains harboring deletions in these pathways resulted in altered drug interaction outcomes (Chevereau & Bollenbach, 2015). This further affirms the validity and significance of the DIR genes identified by INDIGO.”

12. Staph data (Fig S13) - include plots for replicate correlation (mentioned in response to review 1).

We thank the reviewer for these valuable comments; we have revised Fig S13 accordingly.

13. Mtb interaction scores used from literature are not quantitative - should be pointed out.

We have revised the legend of Figure 5 accordingly.

14. There is still no reference in main text for some of the Supp Material.

We thank the reviewer for these valuable comments; we have revised the manuscript accordingly.

References:

- Brynildsen MP, Winkler JA, Spina CS, MacDonald IC, Collins JJ (2013) Potentiating antibacterial activity by predictably enhancing endogenous microbial ROS production. *Nature biotechnology* **31**: 160-165
- Cameron DE, Collins JJ (2014) Tunable protein degradation in bacteria. *Nature biotechnology* **32**: 1276-1281
- Dwyer DJ, Belenky PA, Yang JH, MacDonald IC, Martell JD, Takahashi N, Chan CT, Lobritz MA, Braff D, Schwarz EG, Ye JD, Pati M, Vercruyse M, Ralifo PS, Allison KR, Khalil AS, Ting AY, Walker GC, Collins JJ (2014) Antibiotics induce redox-related physiological alterations as part of their lethality. *Proc Natl Acad Sci U S A* **111**: E2100-2109
- Imlay JA (2015) Diagnosing oxidative stress in bacteria: not as easy as you might think. *Current opinion in microbiology* **24**: 124-131
- Lobritz MA, Belenky P, Porter CBM, Gutierrez A, Yang JH, Schwarz EG, Dwyer DJ, Khalil AS, Collins JJ (2015) Antibiotic efficacy is linked to bacterial cellular respiration. *Proceedings of the National Academy of Sciences*
- Ocampo PS, Lázár V, Papp B, Arnoldini M, zur Wiesch PA, Busa-Fekete R, Fekete G, Pál C, Ackermann M, Bonhoeffer S (2014) Antagonism between bacteriostatic and bactericidal antibiotics is prevalent. *Antimicrobial agents and chemotherapy* **58**: 4573-4582
- Singh R, Ramachandran V, Shandil R, Sharma S, Khandelwal S, Karmarkar M, Kumar N, Solapure S, Saralaya R, Nanduri R (2015) In Silico-Based High-Throughput Screen for Discovery of Novel Combinations for Tuberculosis Treatment. *Antimicrobial agents and chemotherapy* **59**: 5664-5674
- Wildenhain J, Spitzer M, Dolma S, Jarvik N, White R, Roy M, Griffiths E, Bellows David S, Wright Gerard D, Tyers M (2015) Prediction of Synergism from Chemical-Genetic Interactions by Machine Learning. *Cell Systems* **1**: 383-395
- Wood K, Nishida S, Sontag ED, Cluzel P (2012) Mechanism-independent method for predicting response to multidrug combinations in bacteria. *Proceedings of the National Academy of Sciences* **109**: 12254-12259
- Yilancioglu K, Weinstein ZB, Meydan C, Akhmetov A, Toprak I, Durmaz A, Iossifov I, Kazan H, Roth FP, Cokol M (2014) Target-independent prediction of drug synergies using only drug lipophilicity. *Journal of chemical information and modeling* **54**: 2286-2293

3rd Editorial Decision

19 April 2016

We are now satisfied with the modifications made and I am pleased to inform you that we will be able to accept your paper for publication pending the following minor points:

- Figure EV2 is not cited in the text. It should probably be cited in the sentence "The top 81 genes accounted for 50%, the top 222 accounted for 75%..." instead of Figure EV3.
- Datasets EV1, EV2, EV3 should be individually called out as such (using the "EV1" nomenclature) in the text at the appropriate locations. as well as in the data availability section, so that they are properly hyperlinked in the HTML.
- Dataset 4 should be renamed 'Computer Code EV1' and called out as such from the text.
- Please include a short legend and explanation in the first cell of each dataset or, alternatively, include a short README plain text file that is zipped together with the respective dataset.
- The numbering of the subsections within Materials & Methods should be removed.

Corresponding Author Name: Sriram Chandrasekaran

Manuscript Number: MSB-15-6777